# Machine Learning Meets Algebraic Combinatorics: A Suite of Datasets Capturing Research-level Conjecturing Ability in Pure Mathematics

Herman Chau [* 1]  Helen Jenne [* 2]  Davis Brown [* 2 3]  Jesse He [2 4]  Mark Raugas [2]  Sara C. Billey [1]
Henry Kvinge [2 1]

## Abstract

With recent dramatic increases in AI system capabilities, there has been growing interest in utilizing machine learning for reasoning-heavy, quantitative tasks, particularly mathematics. While there are many resources capturing mathematics at the high-school, undergraduate, and graduate level, there are far fewer resources available that align with the level of difficulty and open endedness encountered by professional mathematicians working on open problems. To address this, we introduce a new collection of datasets, the *Algebraic Combinatorics Dataset Repository (ACD Repo)*, representing either foundational results or open problems in algebraic combinatorics, a subfield of mathematics that studies discrete structures arising from abstract algebra. Further differentiating our dataset collection is the fact that it aims at the conjecturing process. Each dataset includes an open-ended research level question and a large collection of examples (up to 10M in some cases) from which conjectures should be generated. We describe all nine datasets, the different ways machine learning models can be applied to them (e.g., training with narrow models followed by interpretability analysis or program synthesis with LLMs), and discuss some of the challenges involved in designing datasets like these.

## 1. Introduction

Modern approaches to machine learning (ML) have been shown to be capable of extracting sophisticated patterns from large and complex datasets. At the same time, there is now increasing evidence that frontier AI systems are capable of performing tasks requiring high-level reasoning capabilities. These trends have led to excitement around the use of machine learning in mathematics. Much of this research explores the use of LLMs and related models to aid in proof writing and mathematical formalization (Song et al., 2024; Yang et al., 2024). While this is an essential part of the mathematician's workflow, there is also a need for machine assisted conjecture generation using (what we call) 'raw' mathematical data. Before identifying a claim that they want to try to prove, a mathematician needs to work through many examples to build intuition and better understand their object of study. For example, when trying to better understand the coefficients of a particular family of polynomials (e.g., Kazhdan-Lusztig polynomials in Section 4), a mathematician may search through countless examples, looking for patterns or other features of interest that may be the basis of future theorems.

Existing applications of machine learning to raw mathematics data tend to fall into several clusters. The first are toy problems (for which we already know many solutions), which are used by the interpretability and science of deep learning communities as a stand-in for more complicated real-world tasks (Zhong et al., 2024; Nanda et al., 2023; Liu et al., 2023; Lee et al.). Another group uses reinforcement learning methods to search for counterexamples to conjectures (Charton et al., 2024; Mehrabian et al., 2024; Wagner, 2021). There is also a growing body of work coming from the mathematics community where off-the-shelf ML methods are just one of several tools used to make progress on a specific problem (Coates et al., 2024; Wagner, 2021; Bao et al., 2022; Kazalicki & Vlah, 2023; Davies et al., 2021). Finally in a few instances, foundation models have begun to be deployed to address specific mathematical questions (Romera-Paredes et al., 2024).

While these works either present interesting methodological progress in ML or valuable results in mathematics, none aim to provide a range of datasets accessible to the broader ML community that represent open or equivalently challenging research-level problems. To fill this gap, we present the

---

[*]Equal contribution  [1]University of Washington  [2]Pacific Northwest National Laboratory  [3]University of Pennsylvania  [4]University of California, San Diego. Correspondence to: Herman Chau <hchau@uw.edu>, Henry Kvinge <henry.kvinge@pnnl.gov>.

*Proceedings of the 42$^{nd}$ International Conference on Machine Learning*, Vancouver, Canada. PMLR 267, 2025. Copyright 2025 by the author(s).

*Algebraic Combinatorics Dataset Repository (ACD Repo)*[1], a collection of 9 datasets consisting of many examples along with an associated question(s). Our collection includes both open problems (e.g., the combinatorial interpretation of Schubert polynomial structure constants) and classic problems whose solution is a major result in the field (e.g., a combinatorial method of calculating the characters of irreducible symmetric group representations).

We choose to restrict ourselves to algebraic combinatorics (an area of mathematics that studies discrete structures arising from abstract algebra) because (i) it requires less background theory to understand, making it generally more accessible to a broader range of researchers, (ii) there already exist specialized software libraries (e.g., Sage (Stein et al., 2024)) designed to efficiently compute many quantities of interest in algebraic combinatorics, and (iii) by nature of being discrete, the objects of interest in algebraic combinatorics tend to be more amendable to representation on a computer.

Each dataset has both an open-ended mathematical question and a related ML friendly task associated to it. The idea is that a model that can effectively solve the ML task has probably learned information that could offer insight into the broader mathematical question. For example, the open question may be finding a combinatorial interpretation of Schubert polynomial structure constants (Section 4.6), which are indexed by triples of permutations. In this case the ML task is to predict the structure constant from the three permutations. For each dataset, we provide context and motivation for the problem, the basic statistics of the dataset, as well as the performance of some basic off-the-shelf models.

We note that these datasets are not designed to be benchmarks in the traditional sense. High performance in terms of standard metrics such as accuracy may be of little value if one is unable to extract a mathematical insight that leads to a fruitful conjecture. In Section 5 we give two examples illustrating how this might be done: (i) by performing a careful interpretability analysis of a performant narrow model (Davies et al., 2021; He et al., 2024) or (ii) by using a LLM that can communicate its reasoning via (for example) code (Austin et al., 2021; Romera-Paredes et al., 2024; Novikov et al., 2025). We hope that these datasets will enable the development of even more effective approaches in the future.

## 2. The Cast of Characters: Partitions, Permutations, and Partial Orders

The field of combinatorics studies a broad range of problems in mathematics centered around discrete objects (e.g, partial

[1] https://github.com/pnnl/ML4AlgComb and https://huggingface.co/ACDRepo

orders, graphs, permutations, partitions) (Stanley, 2011a;b). Ideas and tools from combinatorics play an essential role in many other fields of mathematics and continue to have a strong impact on computer science and physics. Algebraic combinatorics is a subfield of combinatorics that applies combinatorial methods to problems arising from abstract algebra, particularly representation theory and algebraic geometry.

**Partitions:** We use the word partition in this work to mean an integer partition. An *integer partition of* $n \in \mathbb{N}$ is a sequence of positive integers $(n_1, n_2, \ldots, n_k)$ such that $n = n_1 + n_2 + \cdots + n_k$ and $n_1 \geq n_2 \geq \cdots \geq n_k$. We use the standard notation $\mu \vdash n$ to denote that $\mu$ is a partition of $n$. A partition $(n_1, \ldots, n_k)$ is often visualized as a *Young diagram*, with (in English notation) $n_1$ left justified square cells in the first row, $n_2$ left justified square cells in the second row, etc. See Figure 1 (left) for an example of a Young diagram corresponding to the partition $(3, 2, 2)$.

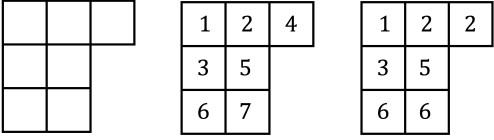

*Figure 1.* **(Left)** A Young diagram for the partition $(3, 2, 2)$. **(Center)** A standard Young tableau for the partition $(3, 2, 2)$. **(Right)** A semistandard Young tableau for the partition $(3, 2, 2)$.

**Young tableaux:** Including extra decorations in the cells in a Young diagram can capture ubiquitous combinatorics found across representation theory and other fields. A *Young tableau* corresponding to a Young diagram $\lambda \vdash n$ is a labeling of the cells of $\lambda$ by an alphabet of symbols. In this work we will consider two types of Young tableaux. A *standard Young tableau* corresponding to partition $\lambda \vdash n$ is a labeling of the cells of $\lambda$ by $1, 2, \ldots, n$ such that the integers strictly increase as one moves down a column or left to right across a row (Figure 1 (center)). The definition of a *semistandard Young tableau* is analogous except that the entries are only assumed to weakly increase as one moves from left to right along a row (see Figure 1 (right)).

**Permutations:** Permutations are familiar in machine learning from their central role in computer science as well as their relevance to symmetries in data and neural networks (Entezari et al.; Ainsworth et al., 2023; Godfrey et al., 2022; Keriven & Peyré, 2019; Zaheer et al., 2017; Lee et al., 2019). There are many ways to represent a permutation. In this paper we use *one-line notation*, which is best illustrated through an example. Suppose $\omega$ is the permutation of the set of elements $\{1, 2, 3, 4\}$ that swaps 1 and 2 and 3 and 4. Then in one-line notation we write $\omega = 2\ 1\ 4\ 3$. 2 is in the first position since 1 is sent to 2, 1 is in the second position

since 2 is sent to 1, 4 is in the third position since 3 is sent to 4, and 3 is in the fourth position since 4 is sent to 3.

Permutations can also be written as sequences of transpositions of adjacent elements. For instance, the permutation $\sigma = 3\ 1\ 2$ can be formed by first swapping 2 and 3 and then the newly adjacent 1 and 3: $1\ 2\ 3 \to 1\ 3\ 2 \to 3\ 1\ 2$. If we denote a transposition of the $i$th and $(i+1)$st element as $s_i$ and read from right to left (as is the convention) then $\sigma$ can be written as $s_1 s_2$. A sequence of adjacent transpositions $s_{i_1} s_{i_2} \dots s_{i_k}$ corresponding to a permutation $\sigma$ is called a *reduced word* if there is no other representation that uses fewer than $k$ adjacent transpositions to represent $\sigma$. Two reduced words are considered *commutation equivalent* if one can be obtained from another by swaps of the form $s_i s_j \to s_j s_i$ where $|i - j| > 1$. Finally, a 3412 *pattern* is a quadruple $(a_i, a_j, a_k, a_\ell)$ such that $i < j < k < \ell$ but $a_k < a_\ell < a_i < a_j$. Patterns have deep connections to algebra and geometry (Billey, 1998).

In the discussion above we implicitly think of permutations of $n$ as bijective functions from $\{1, 2, \dots, n\} \to \{1, 2, \dots, n\}$. Using this perspective, one can define the composition of two permutations. The symmetric group, denoted $S_n$, is defined as the group of permutations on $n$ elements using composition as the group operation. The sequence of transpositions $s_1 s_2$ from the previous paragraph gave an example of the composition of two permutations.

**Posets:** A partially ordered set (poset) is a set $P$ of objects equipped with a binary relation, typically denoted "$\leq$", that is reflexive, antisymmetric, and transitive. This means that for all elements $a, b, c \in P$: (1) $a \leq a$, (2) if $a \leq b$ and $b \leq a$, then $b = a$, and (3) if $a \leq b$ and $b \leq c$, then $a \leq c$. Unlike total orders which are more familiar (e.g., $\mathbb{Z}$), in a partial order some pairs of elements may be incomparable. An example of a partially ordered set is the set of all subsets of $\{1, 2, 3, 4\}$, ordered by inclusion. This is a partial order and not a total order because $\{1, 2\}$ is not comparable to $\{2, 3\}$ or to $\{2, 3, 4\}$, for example. In a poset, $y$ *covers* $x$ if $y$ is greater than $x$ with respect to the ordering, and for any $z$ such that $x \leq z \leq y$, either $z = x$ or $z = y$. In this example, $\{1, 2, 4\}$ covers $\{1, 2\}, \{2, 4\}$, and $\{1, 4\}$, but not $\{1\}, \{2\}$, or $\{4\}$.

## 3. Related Work

**AI for Mathematics:** There is a growing body of work that uses machine learning based methods to assist in mathematics research. With the growing popularity of theorem proving languages such as Lean (Moura & Ullrich, 2021) or Coq (Coq development team, 2004), many of these papers focus more on the proof-creation part of the mathematician's workflow (Song et al., 2024; Yang et al., 2024; Azerbayev et al.). Others aim for more structured, text-based question

and answer in non-research settings (Saxton et al., 2019). In this work we look at large volumes of the raw mathematical data associated to research-level problems. Other work in this vein includes the search for counterexamples in graph theory (Wagner, 2021), the search for connections between different knot invariants (Davies et al., 2021), the classification of $\mathbb{Q}$-Fano varieties (Coates et al., 2024), and Clifford invariants of ADE Coxeter elements (Chen et al., 2024). Unlike these works which aim to shed light on specific problems, this paper's goal is to introduce datasets so that both the expert and non-expert can explore the use of machine learning for research-level mathematics problems.

**Neural Algorithmic Reasoning:** This new field of machine learning looks at applying machine learning methods to algorithmic data (Veličković, 2023). Like mathematics, applying machine learning to algorithmic data provides a setting where arbitrarily large amounts of data can be generated. Further, as with mathematics, working with algorithmic data allows us to manipulate certain aspects of the problem (e.g., complexity) in ways not possible in noisy real-world data. The differences between our work and the primary neural algorithmic reasoning benchmark (Veličković et al., 2022), are substantial. Most notably, the focus of these datasets is not on scientific discovery like the ACD Repo.

## 4. Dataset Descriptions

Here we describe the datasets that are currently included in the ACD Repo. Further details, including dataset statistics, additional problem context, and the methods used for generating the datasets can be found in Appendix B.

Most approaches that use ML to accelerate mathematics research require one to be able to train a model that in some sense understands how to solve a version of the problem. The second (and usually harder) step is to then extract mathematical insights from this model. We aim to give some sense of the difficulty of step one by providing some initial baselines for off-the-shelf models: both narrow (Tables 1, 3, and 4) and LLMs (Table 2 and Table 6). The details of training and testing can be found in Appendix B.10. When training narrow models, we use vanilla architectures as most readers will have some intuition and familiarity with these. It is almost certainly the case that one can do much better by refining model architecture, training routine, data representation, and task structure to better fit the problem. We leave this to users of the datasets.

### 4.1. Computing Characters of Irreducible Representations of the Symmetric Group (Foundational Result)

One way to understand the algebraic structure of permutations (symmetric groups, $S_n$) is through their representation

| Dataset | Logistic regression | MLP | Transformer | Guessing largest class |
|---|---|---|---|---|
| Lattice paths | | | | |
| $n = 10$ | 66.2% | 90.6% ± 0.8% | 65.3% ± 0.0% | 66.2% |
| $n = 11$ | 66.3% | 95.8% ± 0.3% | 69.4% ± 6.0% | 66.3% |
| $n = 12$ | 66.5% | 98.6 % ± 0.1% | 86.2% ± 14.2% | 66.5% |
| Weaving patterns | | | | |
| $n = 6$ | 70.4% | 86.1 % ± 0.2% | 85.9% ± 2.3% | 63.3% |
| $n = 7$ | 85.8% | 99.3 % ± 0.2% | 99.9 % ± 0.4% | 85.0% |
| Cluster algebra quivers | 40.3 % | 86.5 % ± 1.9% | 92.9% ± 0.5% | 17.7% |
| Grassmanian cluster algebras | | | | |
| $n = 6$ | 65.7% | 99.3 % ± 0.1% | 99.5 % ± 0.1% | 50.0% |
| Schubert polynomials | | | | |
| $n = 4$ | 88.8% | 93.1 % ± 2.6% | 94.6% ± 1.0% | 52.3% |
| $n = 5$ | 90.6% | 97.5 % ± 0.2% | 96.2% ± 1.1% | 49.9% |
| $n = 6$ | 89.7% | 99.8 % ± 0.0% | 91.3% ± 8.0% | 50.1% |
| mHeight | | | | |
| $n = 8$ | 91.4% | 99.4 % ± 0.3% | 99.7% ± 0.4% | 91.4% |
| $n = 9$ | 93.2% | 99.8 % ± 0.6% | 99.9% ± 0.4% | 93.2% |
| $n = 10$ | 94.2% | 99.9 % ± 0.0% | 99.9% ± 0.6% | 94.2% |

*Table 1.* Off-the-shelf model accuracy on classification datasets. Results are averaged over three random weight initializations with 95% confidence intervals after a hyperparameter search outlined in Table B.10

theory (Sagan, 2013), which converts abstract algebraic questions into linear algebra questions that are often easier to solve. A *representation* of group $G$ on vector space $V$, is a map $\phi : G \to GL(V)$ that converts elements of $G$ to invertible linear maps from $V$ to $V$ and respects the compositional structure of the group. A basic result in representation theory says that all representations of a finite group can be decomposed into atomic subspace building blocks called *irreducible representations*. Amazingly, irreducible representations are themselves uniquely determined by the value of the trace on matrices, $\phi(g)$, where $g$ ranges over subsets of $G$ called conjugacy classes. These values are called *characters*.

The representation theory of symmetric groups has rich combinatorial interpretations. Both the irreducible representations and the conjugacy classes of $S_n$ are indexed by partitions of $n$ and thus the characters of irreducible representations of $S_n$ are indexed by two partitions of $n$. For $\lambda, \mu \vdash n$ we write $\chi^\lambda_\mu$. This combinatorial connection is not superficial; there are algorithms (e.g., the Murnaghan-Nakayama rule (Stanley, 2011b)), which allow calculation of irreducible characters via simple manipulation of the Young diagrams for $\lambda$ and $\mu$ without any reference to more abstract algebraic structure. We provide datasets for $n = 18, 20, 22$.

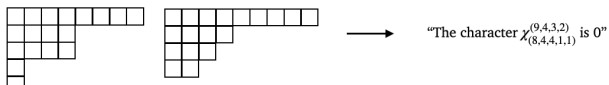

"The character $\chi^{(9,4,3,2)}_{(8,4,4,1,1)}$ is 0"

**ML task:** Train a model that can take two partitions of $n$, $\lambda$ and $\mu$, and predict the corresponding irreducible symmetric group character $\chi^\lambda_\mu$. Does this model learn a known algorithm? If it does not, what method does the model learn? Any novel approaches to computing irreducible symmetric group characters would be of high interest to the mathematics community. This is framed as a regression task.

**Input/output:** Two integer partitions $\mapsto$ an integer.

**How hard is it?:** As can be seen in Table 3, narrow models struggle on this task. This may relate to the long-tail distribution of characters (see Figures 3-5) or the complexity of the problem (Ikenmeyer et al., 2024).

### 4.2. The mHeight Function of a Permutation (Key Construction)

Truly challenging open problems in mathematics often require the development of new mathematical constructions (or even entire new areas of mathematics). This dataset represents a modest example of this. The mHeight function is a statistic associated with a permutation that relates to all 3412-patterns in the permutation. It was developed and plays a crucial role in the proof by Gaetz and Gao (Gaetz & Gao, 2024) which resolved a long-standing conjecture of Billey and Postnikov (Billey & Postnikov, 2005) about the coefficients on Kazhdan-Lusztig polynomials (Section 4.4). The task of predicting the mHeight function represents an interesting opportunity to understand whether a non-trivial intermediate step in an important proof can be learned by machine learning.

Let $\sigma = a_1 \ldots a_n \in S_n$ be a permutation containing at least one occurrence of a 3412 pattern. Let $(a_i, a_j, a_k, a_\ell)$ be a 3412 pattern so that $i < j < k < \ell$ but $a_k < a_\ell < a_i < a_j$ (see Section 2 for more details). The *height* of $(a_i, a_j, a_k, a_\ell)$ is $a_i - a_\ell$. The *mHeight* of $\sigma$ is then the minimum height over all 3412 patterns in $\sigma$. We provide datasets for $n = 8, 9, 10$.

$$\sigma = (6\ 7\ 5\ 4\ 2\ 1\ 3) \longrightarrow \text{"The mHeight function is 2"}$$

**ML task:** Predict the mHeight of a permutation. Since mHeight can take a limited number of values for small $n$, this is framed as a classification problem.

**Input/output:** One permutation $\mapsto$ one of several possible integers.

**How hard is it?:** Both LLMs (Table 2) and narrow models (Table 1) achieve high accuracy (though note that the dataset is heavily imbalanced for larger $n$). Understanding whether the models actually learn some approximation of mHeight or exploit other correlations remains to be studied.

### 4.3. Grassmannian Cluster Algebras and Semistandard Young Tableaux (Open Problem)

The Grassmann manifold $\text{Gr}(k, n)$ is the set of full-rank $k \times n$ matrices up to equivalence of elementary row operations (equivalently the space whose points are $k$-dimensional subspaces in $\mathbb{R}^n$). Grassmannians are of fundamental geometric importance and are a central tool in a model of quantum field theory known as supersymmetric Yang-Mills theory (Golden et al., 2014).

Among the many algebraic-combinatorial properties of Grassmannians is an algebraic structure on its coordinate ring making it something called a cluster algebra (Scott, 2006; Williams, 2014). A recent result of Chang, Duan, Fraser, and Li (Chang et al., 2020) parameterize cluster variables of the Grassmannian coordinate ring in terms of equivalence classes of semistandard Young tableaux (SSYTs). Not every SSYT indexes a cluster variable and a natural question to ask is which are valid cluster variable indices. A necessary condition is that the tableau is of rectangular shape. We follow the set-up of (Cheung et al., 2023) who first applied machine learning to this problem, though we choose a different method of sampling tableaux that do not index cluster variables. The math question is thus to find a concise combinatorial characterization of those SSYT that index a cluster variable. We provide a dataset for $\text{Gr}(3, 12)$, restricting to rank 4. This corresponds to rectangular SSYT of shape $3 \times 4$ with entries drawn from $\{1, 2, \ldots, 12\}$.

| 2 | 3 | 4 | 7 |
|---|---|---|---|
| 3 | 5 | 6 | 8 |
| 6 | 9 | 11 | 12 |

$\longrightarrow$ "This semistandard Young tableau indexes a cluster variable"

| Dataset | o1-Mini | GPT-4o Mini | GPT-4o |
|---|---|---|---|
| In-context learning | | | |
| $n = 10$ | 95.4% | 89.5% | 95.5% |
| $n = 11$ | 97.0% | 97.0% | 97.0% |
| Program synthesis | | | |
| $n = 10$ | 94.2% | 94.2% | 94.2% |
| $n = 11$ | 95.5% | 95.5% | 95.5% |

*Table 2.* Success of GPT-4o-mini and GPT-4o solving the mHeight function task via either in-context learning or program synthesis. Hyperparameters for these experiments can be found in Section B.11.

**ML task:** Predict whether a Young tableau indexes a cluster variable. This is a binary classification task.

**Input/output:** A SSYT of shape $3 \times 4$ with entries drawn from $\{1, 2, \ldots, 12\} \mapsto$ Boolean indicating whether the tableau indexes a cluster variable or not.

**How hard is it?:** Even naive methods can result in high accuracy (Table 1), suggesting that there is something interesting here to learn.

### 4.4. The Coefficients of Kazhdan-Lusztig Polynomials (Open Problem)

Kazhdan-Lusztig (KL) polynomials are integer polynomials in a variable $q$ that (for our purposes) are indexed by a pair of permutations (Kazhdan & Lusztig, 1979). We will write the KL polynomial associated with permutations $x$ and $w$ as $P_{x,w}(q)$. For example, the KL polynomial associated with permutations $x = 1\ 4\ 3\ 2\ 7\ 6\ 5\ 10\ 9\ 8\ 11$ and $w = 4\ 6\ 7\ 8\ 9\ 10\ 1\ 11\ 2\ 3\ 5$ is

$$P_{x,w}(q) = 1 + 16q + 103q^2 + 337q^3 + 566q^4 + 529q^5 + 275q^6 + 66q^7 + 3q^8$$

(this example was computed by (Warrington)). KL polynomials have deep connections throughout several areas of mathematics. For example, KL polynomials are related to the dimensions of intersection homology in Schubert calculus and the representation theory of the Hecke algebra. They can be computed via a recursive formula (Kazhdan & Lusztig, 1979), nevertheless, in many ways they remain mysterious. For instance, there is no known closed formula for the degree of $P_{x,w}(q)$.

Mathematicians would like a better understanding of the coefficients on powers of $q$ in $P_{x,w}(q)$. For example, one question concerns the coefficient on term $q^{\ell(x)-\ell(w)-1/2}$ (where $\ell(x)$ is a statistic called the length of the permutation), which is known as the $\mu$-coefficient. Both the constant term and coefficient on $q$ are well-understood. We provide datasets for $n = 5, 6, 7$.

$$\begin{aligned} x &= (1\ 3\ 2\ 4\ 5\ 7\ 6\ 8) \\ w &= (3\ 4\ 1\ 2\ 7\ 8\ 5\ 6) \end{aligned} \longrightarrow \text{"In } P_{x,w}(q) \text{ the coefficient on } q^2 \text{ is } 1\text{"}$$

**ML Tasks:** Predict one or more coefficients of $P_{x,w}(q)$

given $x$ and $w$. Empirically, for fixed $n$ and power of $q$, coefficients tend to only take a few values so we frame this as a classification task.

**Input/output:** Two permutations $\mapsto$ one of several integers.

**How hard is it?:** We provide both accuracy (Table 4) and F1-scores (Table 5) for these imbalanced datasets. Narrow models perform well.

### 4.5. The Robinson-Schensted-Knuth Correspondence (Foundational Result)

The Robinson-Schensted-Knuth (RSK) algorithm (Robinson, 1938; Schensted, 1961) gives a bijection between pairs of semistandard Young tableaux of the same shape and matrices with non-negative integer entries. The special case we consider (which is sometimes called the Robinson-Schensted algorithm) restricts to a bijection between pairs of standard Young tableaux and permutations in $S_n$. This correspondence is significant in algebraic combinatorics because it connects two of the most fundamental objects in the field (see (Stanley, 1984) for additional history and context).

Given its fundamental importance, it would be interesting to see whether a model learns the RSK algorithm given enough examples of the correspondence. To this end the dataset consists of triples: two standard Young tableaux of size $n$ and their corresponding permutation (obtained via the RSK algorithm). We provide datasets for $n = 8, 9$.

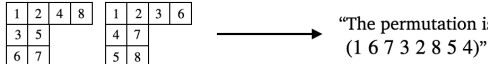

"The permutation is (1 6 7 3 2 8 5 4)"

**ML task:** Given a pair of standard Young tableaux, predict the corresponding permutation. The task is framed as regression and for the permutation target we use a vector representing its descent set (we found this to be an easier target for regression than representing the permutation in 1-line notation).

**Input/output:** Two standard Young tableaux of the same shape $\mapsto$ a permutation.

**How hard is it?:** None of the small, narrow models learned this task in our experiments, achieving results similar or worse than simply guessing the mean of all permutations (Table 3). Like symmetric group character calculation, the RSK algorithm has intricate combinatorial rules that may require larger and more capable models or more elaborate training strategies.

### 4.6. Schubert Polynomial Structure Constants (Open Problem)

Schubert polynomials (Bernstein et al., 1973; Demazure, 1974; Lascoux & Schützenberger, 1982) are a family of

polynomials indexed by permutations of $S_n$. Developed to study the cohomology ring of the flag variety, they have deep connections to algebraic geometry, Lie theory, and representation theory. Despite their geometric origins, Schubert polynomials can be described completely combinatorially (Billey et al., 1993; Bergeron & Billey, 1993), making them a well-studied object in algebraic combinatorics. An important open problem in the study of Schubert polynomials is understanding their *structure constants*. When two Schubert polynomials are multiplied, their product is a linear combination of Schubert polynomials, $\mathfrak{S}_\alpha \mathfrak{S}_\beta = \sum_\gamma c_{\alpha\beta}^\gamma \mathfrak{S}_\gamma$. The $c_{\alpha\beta}^\gamma$ are conjectured to have a combinatorial description or formula (most likely related to permutations $\alpha$, $\beta$, and $\gamma$). To give an example of what we mean by combinatorial description, the structure constants of Schur polynomials (a special type of Schubert polynomial) count the number of semistandard tableaux satisfying certain properties.

Each instance in this dataset is a triple of permutations $(\alpha, \beta, \gamma)$, labeled by its coefficient $c_{\alpha\beta}^\gamma$ in the expansion of the product $\mathfrak{S}_\alpha \mathfrak{S}_\beta$. Not all possible triples of permutations are included; the dataset consists of an approximately equal number of zero and nonzero coefficients. We provide datasets for $n = 4, 5, 6$.

$$\sigma = (1\ 2\ 4\ 3), \quad \nu = (1\ 4\ 3\ 2), \quad \longrightarrow \quad \text{"The structure constant } c_{\sigma,\nu}^\mu$$
$$\mu = (1\ 5\ 3\ 2\ 4) \qquad\qquad \text{on } S_\mu \text{ in } S_\sigma \star S_\nu \text{ is 1"}$$

**ML task:** Train a model that given three permutations $\alpha, \beta, \gamma$, can predict the associated structure constant $c_{\alpha,\beta}^\gamma$. Since these structure constants only take a few values for small permutations, we frame this as a classification task.

**Input/output:** Three permutations $\mapsto$ one of several integers.

**How hard is it?:** Both small MLPs and transformers can achieve high accuracy (Table 1) as well as LLMs via program synthesis (Table 6). Some of the latter is an artifact of how we originally sampled zero-valued structure constants.

### 4.7. Partial Orders on Lattice Paths (Open Problem)

Consider northeast lattice paths that travel along the edges of a grid from $(0,0)$ to $(a, b)$, only taking steps north and east and never passing through the diagonal $y = \frac{b}{a}x$, where $a$ and $b$ are relatively prime. (Schiffler, 2023) defines two order relations on such paths called the *matching ordering* ($\leq_M$) and the *Lagrange ordering* ($\leq_L$), motivated by questions in number theory. The matching ordering assigns a number to each lattice path based on the number of perfect matchings of an associated snake graph, while the Lagrange ordering assigns a number to each lattice path equal to the Lagrange number of a certain continued fraction. These numbers each define the respective partial order. Mathematicians would be interested to better understand the rela-

tionship between these orders (Apruzzese & Cong, 2023). We provide datasets for lattice paths from zero to $(10, 9)$, $(11, 10)$, $(12, 11)$, and $(13, 12)$.

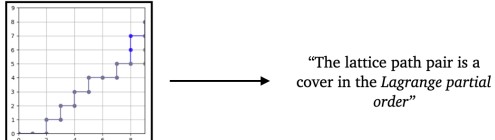

"The lattice path pair is a cover in the *Lagrange partial order*"

**ML task:** Given a pair of lattice paths $(w, w')$, train a model that can predict whether $w'$ covers $w$ (see Section 2) in either the matching or Lagrange order.

**Input/output:** Two lattice paths $\mapsto$ whether the pair is a cover relation in matching or Lagrange order.

**How hard is it?:** MLPs achieve good performance (Table 1). On the other hand, we found it challenging to train performant transformers.

### 4.8. Mutation Equivalence of Quivers (Open Problem)

Quivers and quiver mutations are central to the combinatorial study of cluster algebras, algebraic structures with connections to Poisson Geometry, string theory, and Teichmuller theory. Quivers are directed (multi)graphs, and a quiver mutation is a local transformation centered at a chosen node of the graph that involves adding, deleting and reversing the orientation of specific edges based on a set of combinatorial rules. A fundamental open problem in this area is finding an algorithm that determines whether two quivers are mutation equivalent (one can traverse from one quiver to another by applying mutations). Currently, such algorithms only exist for special cases, including types $A$ (Buan & Vatne, 2008), $D$ (Vatne, 2010), and $\tilde{B}$, $\tilde{C}$, and $\tilde{D}$ (Henrich, 2011). The $\tilde{B}$ and $\tilde{C}$ types correspond to the classes $BD$ and $BB$ in our dataset, respectively. Consistent with Sage we use the naive notation, which specifies a quiver by indicating the two ends of the diagram, which are joined by a path (Musiker & Stump, 2011). To our knowledge, the remaining classes in this dataset ($E$, $DE$, $BE$) lack characterizations. Recent work has explored whether deep learning models can learn to correctly predict if two quivers are mutation equivalent (Bao et al., 2020). (He et al., 2024) utilized an alternative version of this dataset to re-discover known characterization theorems. The dataset consists of adjacency matrices for quivers drawn from 7 different mutation equivalence classes ($A$, $D$, $E$, $DE$, $BE$, $BD$, and $BB$).

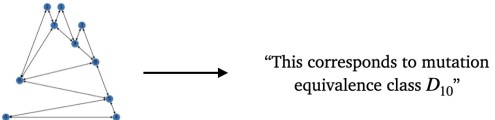

"This corresponds to mutation equivalence class $D_{10}$"

**Task:** Find simple rules to determine which of 7 different mutation equivalence classes a quiver belongs to. This is framed as a classification task.

**Input/output:** An adjacency matrix $\mapsto$ one of 7 different possible labels.

**How hard is it?:** Transformers and MLPs achieve reasonable accuracy on this task ( Table 1). (He et al., 2024) was able to train a far more performant model (accuracy $> 99\%$) and re-discover several known characterization theorems using a specialized graph neural network architecure.

### 4.9. Weaving Patterns (Open Problem)

Weaving patterns are $n \times n - 1$-matrices with $\{1, 2, \ldots, n\}$-entries introduced by Felsner (1997) to study the number of reduced decompositions of the permutation $\sigma = n \ n - 1 \ldots 1$ up to commutation equivalence. The number of such objects also counts the number of parallel sorting networks, the number of rhombic tilings of regular polygons, and is connected to the study of the higher Bruhat orders (Chau, 2024). An $O(n^2)$ algorithm for determining if a given $\{1, 2, \ldots, n\}$-matrix is a valid weaving pattern exists but gives no additional insight into the structure of weaving patterns and correspondingly the asymptotics of reduced decompositions.

The enumeration of reduced decompositions up to commutation equivalence has been studied by many including Knuth and Stanley. An exact formula is likely out of reach, so asymptotic upper and lower bounds are of great interest. ML models that can detect necessary or sufficient conditions for a matrix to be a valid weaving pattern have the potential to lead to substantial improvements in the upper bound.

Each dataset is a mixture of enriched weaving patterns and non-weaving pattern matrices with $\{1, 2, \ldots, n\}$-entries. We provide datasets for $n = 6, 7$.

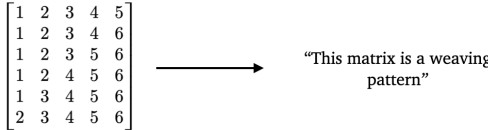

"This matrix is a weaving pattern"

**ML task:** Train a model to classify whether a $\{1, 2, \ldots, n\}$-matrix is a weaving pattern or not. This task is framed as binary classification.

**Input/output:** An $n \times n - 1$, $\{1, 2, \ldots, n\}$-matrix $\mapsto$ binary label.

**How hard is it?:** Especially in the $n = 7$ regime, even small MLPs and transformers achieve high accuracy (Table 1).

## 5. Case Studies of Applying ML to ACD Repo Datasets

We begin this section by providing two examples for how conjectures can be generated using a combination of tools from machine learning and an expert-in-the-loop. Both utilize a dataset (or a derivative of a dataset) from the ACD Repo. Our purpose is not to propose novel ML methods nor to describe novel mathematics, but rather to provide the reader with prototype examples of how the ACD Repo can be used in practice. For a canonical instance where machine learning assisted conjectured generation was used to prove new mathematics see (Davies et al., 2021) (for the conjecture) and (Blundell et al., 2022) (for the theorem).

**Graph Neural Networks, XAI, and Quiver Mutation Equivalence:** (Davies et al., 2021) was one of the first works to show that mathematical conjectures can be made through careful analysis of a machine learning model trained to solve a task around an open problem in mathematics. The basic outline is (1) train a model on the ML friendly task related to the open problem of interest, (2) analyze the resulting performant model to understand how it is making its predictions, and (3) generate conjectures based on this analysis.

(He et al., 2024) applied this approach to a modified version of the quiver mutation equivalence dataset from the ACD Repo (Section 4.8). They showed that they could re-discover characterization theorems for type $D$ and type $\tilde{D}$ quivers. To do this, they trained a novel variant of a message passing graph neural network specifically designed to be able to capture the presence of subgraphs (which have been used to determine membership to several other notable mutation equivalence classes). Having trained a model that achieves high accuracy, the authors used latent space clustering within classes to break down the problem into distinct cases. Then they used a graph neural network explainability technique called PGExplainer (Luo et al., 2020) to identify the subgraph motifs that characterize each of these clusters. In the end they re-discovered theorems from (Vatne, 2010) and (Henrich, 2011).

**Program Synthesis for Schubert Polynomials:** The previous example highlighted the use of narrow models for conjecture generation. In our second example we describe the use of foundation models for a less labor- but more compute-intensive version of conjecture generation. Conjecture generation is a crucial component in current evolutionary approaches to mathematical discovery, notably (Romera-Paredes et al., 2024; Novikov et al., 2025) (though note that there are other non-LLM-based approaches to conjecture generation (Gauthier & Urban, 2023; Raayoni et al., 2021; Fajtlowicz, 1988)). The basic idea is that because computer code is more interpretable than raw ML predictions, one can use an LLM to generate (human-interpretable)

code that solves a mathematical task. This is often known as *program synthesis*.

We applied this procedure to the Schubert polynomial structure constants dataset. There are various additional steps that one can use in program synthesis to improve code generation (for instance, in (Romera-Paredes et al., 2024) the authors interleave code generation with the application of an evolutionary algorithm). For our experiments we only performed a single round of 100 generated Python programs with no additional steps. We were surprised to find that (if we provided the proper mathematical context in our prompt) o1-mini, Claude, and sometimes 4o were able to produce Python programs that achieved 100% accuracy on the test set. In originally generating this dataset, we subsampled from the set of zero valued structure constants because these are far more common than non-zero valued structure constants. Interestingly, examination showed that in some cases the LLMs had reversed engineered our sampling algorithm, which essentially involved applying a single transposition to one of the three permutations indexing a non-zero structure constant to obtain a zero structure constant.

It turns out that in the cases we applied program synthesis to $(n = 4, 5)$, this process introduces a spurious correlation in the training and evaluation data[2]: structure constant $c_{\alpha,\beta}^{\gamma} \neq 0$ if and only if $\ell(\alpha) + \ell(\beta) + \ell(\gamma) \neq 0 \mod 2$ where $\ell$ is a statistic on permutations called the length. We note that the length statistic is not included in the dataset itself or in the prompt we gave the model to initiate program synthesis.

Though this exercise did not provide any useful mathematical insights, it does show that the LLM-based approach to conjecture generation carries the benefit that in some cases the LLM can exploit background tools and concepts (such as the length of a permutation) not available to narrow models. More speculatively, it may be that program synthesis tends to elicit such background information to a greater extent than other methods. For comparison, asking these models to solve this task directly using in-context learning, resulted in only 50-60% accuracy.

## 6. Discussion

**Narrow Model Performance:** We trained small MLPs, decoder only transformers, and logistic/linear regression models on each dataset in the ACD Repo and measured their test performance using either accuracy (classification tasks) or mean squared error (regression tasks). The goal of this exercise was to approximately measure the relative difficulty of training a performant model (though note that a performant model alone does not translate to mathematical discovery). We found that MLPs tended to perform most consistently across tasks and therefore might be a good

---

[2]This issue has since been fixed.

place to start when exploring the use of smaller models. Transformers also performed well but struggled in a few cases (such as the lattice paths dataset).

Though we cannot prove why some dataset/architecture pairs failed to perform well (especially for datasets corresponding to open problems), several factors should be considered. Firstly, the loss landscape for tasks involving discrete structures like permutations seems to be less favorable for architectures and training approaches that we currently use in deep learning. For example, many properties in combinatorics are highly sensitive to small changes to the input (e.g., the parity of a permutation). Such tasks may be intrinsically challenging within current frameworks (Hahn & Rofin, 2024).

Further, the representation of input data can have a large impact on model performance. For instance, we found it challenging to train a narrow model (MLP or transformer) to predict the parity of a permutation represented in one-line notation. But this was feasible when permutations were represented via their inversion vector. While effective representations can sometimes be found when a solution is already known, this strategy is not available for open problems. When designing these datasets we chose not to optimize input or output representation, leaving this to users.

Unsurprisingly, we found that larger datasets were generally associated with better model performance. This is true even in the case where generating a larger dataset required increasing $n$, and thus making the problem potentially more complex (e.g., working with partitions of $n + 1$ rather than partitions of $n$). There were some tasks however that seem hard even when the dataset size is increased. As can be seen in Table 3, performance regressing symmetric group characters is very poor. This may relate to the complexity of the task (calculating symmetric group characters is known to belong to $\#P$ (Hepler, 1996)) or to the distribution of symmetric group characters which has a very long tail.

**Challenges in the generation of mathematics datasets:** Imbalance is an issue in several different respects. On the one hand, traditional class imbalance comes up frequently and can be quite extreme for large values of $n$ where asymptotic properties begin to take hold. More broadly, the mathematically interesting objects are frequently interesting precisely because they are rare. For a given task, it may be the case that randomly sampled instances will be uninteresting with high probability because they can be predicted or classified for straightforward reasons. One way to mitigate this situation is to subsample for harder examples. This is what we did for several of the datasets in the ACD Repo including Weaving Patterns where we imposed some additional constraints on the non-weaving pattern $\{1, 2, \ldots, n\}$-matrices to make them harder to distinguish from true weaving pat-

terns (but care must be taken so that the model does not simply discover one's sampling procedure).

Unlike other domains where data is more rare, the process of mathematical discovery using machine learning may involve more directly optimization over both a model and dataset. That is, modifying model and dataset so that the model learns more informative, robust, and effective representations.

Finally, there exist a near limitless number of open problems in mathematics and finding the optimal ones for inclusion was challenging. We aimed to find problems such that (i) some working mathematicians care about the solution, (ii) a significant amount of data (at least $> 10k$ instances, but preferably many more) could easily be generated, and (iii) there is an ML friendly simplification of the problem (this can be challenging with extremely abstract problems for instance).

**A Challenge to the Interpretability Community:** There are now many cases where researchers in mechanistic interpretability have worked out detailed descriptions of the ways in which models of various sizes solve tasks. For example, reverse engineering the ways in which small transformers perform mathematical tasks such as modular arithmetic (Nanda et al., 2023). While such examples may fall short of larger goals such as being able to audit foundation models on fuzzy real-world tasks, they align perfectly with the needs of AI for math where an algorithmic description of the method by which a model has learned to solve an open problem could easily translate into a theorem (or at least a novel algorithm). As such, we propose these datasets as an achievable but high impact opportunity for the field of mechanistic interpretability.

# 7. Conclusion

In this paper we introduce the Algebraic Combinatorics Dataset Repository, a collection of datasets representing open problems or research-level foundational results that have been structured for machine learning. With all the advancements in model capability and model explainability, we believe that machine learning holds the potential to dramatically accelerate progress in mathematics. Signs of this are already starting to appear, e.g. (Novikov et al., 2025). On the other hand, mathematics is a vast field and broad, ML-fueled acceleration across sub-disciplines will require tools that can be used by researchers across a broad range of institutions. We hope that these datasets will fuel the development of these tools.

## Acknowledgements

This research was supported by the Mathematics for Artificial Reasoning in Science (MARS) initiative at Pacific Northwest National Laboratory. It was conducted under the Laboratory Directed Research and Development (LDRD) Program at at Pacific Northwest National Laboratory (PNNL), a multiprogram National Laboratory operated by Battelle Memorial Institute for the U.S. Department of Energy under Contract DE-AC05-76RL01830.

## Impact Statement

This paper presents work whose goal is to simultaneously advance the field of Machine Learning and Mathematics. There are many potential societal consequences of our work, none which we feel must be specifically highlighted here.

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

| Dataset | Linear regression | MLP | Transformer | Guess training label mean |
|---|---|---|---|---|
| $S_n$ characters | | | | |
| $n = 18$ | $1.5920 \times 10^{10}$ | $2.7447 \times 10^9 \pm 8.86015 \times 10^8$ | $2.4913 \times 10^{10} \pm 1.4350 \times 10^7$ | $1.5920 \times 10^{10}$ |
| $n = 20$ | $4.2007 \times 10^{12}$ | $4.2254 \times 10^{12} \pm 5.1236 \times 10^{11}$ | $5.3897 \times 10^{12} \pm 3.6464 \times 10^{11}$ | $4.2007 \times 10^{12}$ |
| $n = 22$ | $8.0395 \times 10^{14}$ | $1.1192 \times 10^{14} \pm 4.9321 \times 10^{12}$ | $1.3797 \times 10^{14} \pm 6.2799 \times 10^{12}$ | $8.0395 \times 10^{14}$ |
| RSK | | | | |
| $n = 8$ | 0.21 | $0.43 \pm 0.05$ | $1.51 \pm 0.02$ | 0.21 |
| $n = 9$ | 0.21 | $0.96 \pm 0.07$ | $3.85 \pm 0.09$ | 0.21 |

*Table 3.* Off-the-shelf model MSE on regression datasets. Results are averaged over three random weight initializations with 95% confidence intervals after a hyperparameter search outlined in Appendix B.10.

| Dataset/coefficient | MLP | Transformer | Guessing largest class |
|---|---|---|---|
| $n = 5$ | | | |
| 1 | $99.8\% \pm 0.2\%$ | $99.9\% \pm 0.1\%$ | 73.7% |
| $q$ | $99.5\% \pm 0.4\%$ | $99.2\% \pm 1.0\%$ | 97.0% |
| $q^2$ | $99.9\% \pm 0.1\%$ | $100.0\% \pm 0.0\%$ | 99.9% |
| $n = 6$ | | | |
| 1 | $99.9\% \pm 0.0\%$ | $100.0\% \pm 0.0\%$ | 80.9% |
| $q$ | $99.9\% \pm 0.0\%$ | $99.9 \pm 0.0$ | 95.8% |
| $q^2$ | $99.9\% \pm 0.0\%$ | $99.9 \pm 0.0$ | 99.5% |
| $q^3$ | $99.9\% \pm 0.0\%$ | $99.9 \pm 0.0$ | 99.9% |

*Table 4.* Baseline model classification accuracy for predicting KL polynomial coefficients for $n = 5, 6$. Results are averaged over three random weight initializations with 95% confidence intervals. The MLPs had layer dimension 256 and depth 4 and were trained with a learning rate of 0.0005. The transformers had dimension 256, depth 6, 8 heads, and were trained with a learning rate of 0.0005.

## A. Dependence on $n$

Many problems in algebraic combinatorics have a natural dependence on a parameter $n$ (e.g., permutations are parameterized by the number of elements that they permute). We have chosen to structure datasets in the ACD Repo to reflect this, with the majority of datasets belonging to a series $\{D_n\}_{n \geq 1}$. We provide a few values of $n$ and, in many cases, the code to generate others. Generally, there are two properties that change as $n \to \infty$. First, the size of $D_n$ grows as $n$ grows. The rate of growth depends on the specific problem, with many $|D_n|$ growing exponentially (such as those datasets that depend on the number of permutations of $n$). On the other hand, the problems can also become harder in various ways as $n$ grows.

Experimentally we have found that larger values of $n$ tend to lead to better model performance regardless of this potential increase in difficulty. For example, we ran 5 2-layer MLP models for 500 epochs on the lattice path datasets corresponding

| Dataset/coefficient | MLP | Transformer | Guessing largest class |
|---|---|---|---|
| $n = 5$ | | | |
| 1 | $99.7\% \pm 0.1\%$ | $99.9\% \pm 0.4\%$ | 73.7% |
| $q$ | $93.9\% \pm 3.7\%$ | $92.7\% \pm 7.6\%$ | 97.0% |
| $q^2$ | $50.0\% \pm 0.0\%$ | $100.0\% \pm 0.0\%$ | 99.9% |
| $n = 6$ | | | |
| 1 | $99.9\% \pm 0.0\%$ | $100.0\% \pm 0.0\%$ | 80.9% |
| $q$ | $99.0\% \pm 1.5\%$ | $98.0 \pm 3.7$ | 95.8% |
| $q^2$ | $97.4\% \pm 5.2\%$ | $86.2 \pm 47.7$ | 99.5% |
| $q^3$ | $87.9\% \pm 4.5\%$ | $88.3 \pm 17.1$ | 99.9% |

*Table 5.* Baseline model classification macro F1-scores (to account for class imbalance) for predicting KL polynomial coefficients for $n = 5, 6$. Results are averaged over three random weight initializations with 95% confidence intervals. The MLPs had layer dimension 256 and depth 4 and were trained with a learning rate of 0.0005. The transformers had dimension 256, depth 6, 8 heads, and were trained with a learning rate of 0.0005.

| Dataset | Claude 3.5 Sonnet | GPT-4o Mini | GPT-4o |
|---|---|---|---|
| In-context learning | | | |
| $n = 3$ | 76.4% | 64.7% | 58.8% |
| $n = 4$ | 59.5% | 53.5% | 57.0% |
| $n = 5$ | 58.5% | 51.5% | 57.0% |
| Program synthesis | | | |
| $n = 3$ | 94.1% | 82.4% | 100.0% |
| $n = 4$ | 65.0% | 100.0% | 100.0% |
| $n = 5$ | 99.8% | 64.2% | 64.7% |

*Table 6.* Success of Claude 3.5 Sonnet, GPT-4o Mini, and GPT-4o solving the Schubert polynomial structure constant task via either in-context learning or program synthesis. Hyperparameters for these experiments can be found in Section B.11. As described in Section 5, models that achieved $100\%$ had effectively learned the strategy by which we subsampled zero-valued structure constants. This has now been fixed.

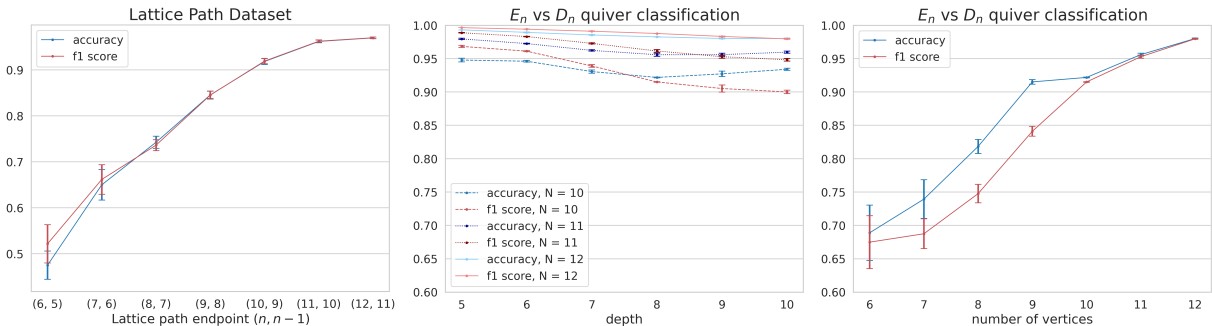

*Figure 2.* (**Left**) Performance on the Lattice Path Dataset as a function of the lattice path endpoint (larger endpoint values means longer and more paths). As $n$ grows in $n \times n - 1$, the training set size increases but the problem may also grow harder. (**Center**) Performance on the type $E$ versus type $D$ quiver classification task as a function of the depth, which must be specified for type $E$ quivers on $n = 10, 11, 12$ vertices, and (**Right**) the number of vertices $n$.

to grids of size $6 \times 5, \dots, 13 \times 12$. We see in Figure 2 (left) that with an interesting exception of moving from $7 \times 6$ to $8 \times 7$, performance across a range of dimensions improves as $n$ grows. There are exceptions, however. We looked at sampling from greater depth when exploring the quiver mutation equivalence dataset (Section 4.8). This effectively means allowing a greater number of mutations to be applied to the initial quiver when generating the dataset. As shown in Figure 2 (center), we find that performance somewhat degrades even though the size of the datasets increase. We suspect that exploration of the complexity of these problems (where it is known) might be an avenue for shedding light on this phenomenon.

## B. Dataset details

All datasets are stored as `.txt` files with one datum instance per line. In this section we will describe each file and explain how to interpret it. Functions capable of loading and parsing each file are available on the GitHub page.

### B.1. Computing Characters of Irreducible Representations of the Symmetric Group

Since the conjugacy classes of the symmetric group $S_n$ are indexed by integer partitions of $n$, characters are constant on conjugacy classes, and the irreducible representations of $S_n$ are also indexed by integer partitions of $n$, the task is to use a pair of integer partitions of $n$ to predict the character of the corresponding irreducible representation of the symmetric group.

Within each file, two integer partitions are provided followed by an integer corresponding to the character. For instance, the line

```
[3,1,1],[2,2,1],-2
```

says that the character $\chi^{3,1,1}_{2,2,1} = -2$.

In all cases the characters are heavily concentrated around 0 with very long tails. This likely contributes to the difficulty of the task and could be overcome with some simple pre- and post-processing. We have not chosen to do this in our baselines.

**Characters of $S_{18}$**

There are $118,580$ training examples and $29,645$ test examples. The maximum character value is $16,336,320$. The minimum character value is $-1,223,040$.

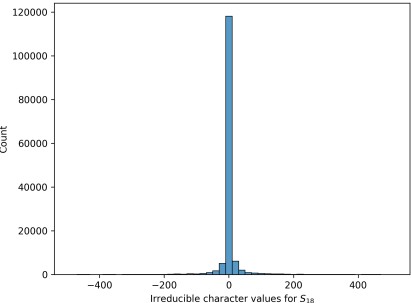

*Figure 3.* Histogram of $S_{18}$ characters within the interval $[-500, 500]$

**Characters of $S_{20}$**

There are $298,661$ training examples and $74,819$ test examples. The maximum character value is $249,420,600$. The minimum character value is $-17,592,960$.

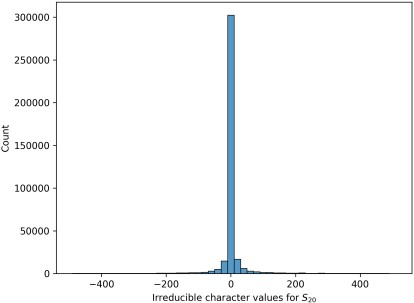

*Figure 4.* Histogram of $S_{20}$ characters within the interval $[-500, 500]$

**Characters of $S_{22}$**

There are $763,109$ training examples and $190,726$ test examples. The maximum character value is $5,462,865,408$. The minimum character value is $-279,734,796$.

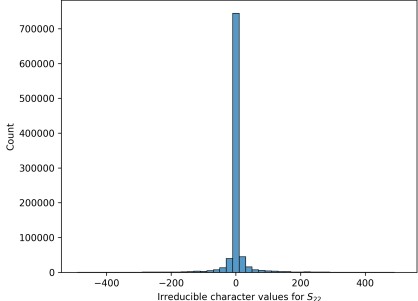

*Figure 5.* Histogram of $S_{22}$ characters within the interval $[-500, 500]$

|              | 0      | 1    | 2   | 3  | 4  |
| ------------ | ------ | ---- | --- | -- | -- |
| Training set | 6, 716 | 508  | 78  | 9  | 1  |
| Testing set  | 1, 672 | 136  | 18  | 3  | 0  |

*Table 7.* Statistics for the mHeight dataset for $n = 8$.

|              | 0       | 1      | 2    | 3   | 4  | 5  |
| ------------ | ------- | ------ | ---- | --- | -- | -- |
| Training set | 49, 092 | 3, 161 | 524  | 77  | 9  | 1  |
| Testing set  | 12, 317 | 759    | 118  | 19  | 3  | 0  |

*Table 8.* Statistics for the mHeight dataset for $n = 9$.

Sage (Stein et al., 2024) was used to calculate character values. Data generation scripts can be found on the GitHub page.

### B.2. The mHeight Function of a Permutation

This dataset contains permutations labeled by their mHeight. An example calculation of mHeight is shown in Figure 6. Permutations are written in 1-line notation followed by the symbol ';' and then the value of the mHeight function on the permutation. So

    (6, 8, 7, 5, 4, 9, 3, 0, 1, 2);1

can be read as saying that the permutation 6 8 7 5 4 9 3 0 1 2 has mHeight 1.

We provide datasets for permutations of size $n = 8, 9, 10$. The files are called:

- `mHeight_8_train.txt`

- `mHeight_8_test.txt`

- `mHeight_9_train.txt`

- `mHeight_9_test.txt`

- `mHeight_10_train.txt`

- `mHeight_10_test.txt`

The dataset was generated using a Python script which can be found on the GitHub page. Dataset statistics can be found in Table 7-Table 9.

**Permutation**
$$\sigma = 3\,4\,5\,1\,2$$
**3 4 1 2 patterns**
3 4 1 2   4 5 1 2   3 5 1 2
**mHeight of $\sigma$**
$$\max(3 - 2, 4 - 2, 3 - 1) = 2$$

*Figure 6.* An example of the calculation of mHeight on a permutation.

|              | 0        | 1       | 2      | 3    | 4   | 5   | 6  |
| ------------ | -------- | ------- | ------ | ---- | --- | --- | -- |
| Training set | 352, 494 | 17, 952 | 3, 079 | 502  | 74  | 10  | 1  |
| Testing set  | 88, 058  | 4, 503  | 803    | 140  | 22  | 2   | 0  |

*Table 9.* Statistics for the mHeight dataset for $n = 10$.

|  | Cluster variable | Not cluster variable | Total |
|---|---|---|---|
| Training set | 74, 329 | 74, 329 | 148, 658 |
| Testing set | 18, 582 | 18, 582 | 37, 164 |

*Table 10.* Statistics of the Grassmannian cluster algebra dataset.

## B.3. Grassmannian Cluster Algebras and Semistandard Young Tableaux

This dataset relates to a cluster algebra associated with the Grassmann manifold $\text{Gr}(k, n)$. Each cluster variable is indexed by a rectangular SSYT with $k$ rows with entries drawn from $\{1, \ldots, n\}$. The *rank* of these rectangular SSYT (and the rank of the associated cluster variable) is given by their number of columns. Following (Cheung et al., 2023), in this dataset we focus on $\text{Gr}(3, 12)$ and hence look at rectangular SSYT with 3 rows filled with entries drawn from $\{1, \ldots, 12\}$. We further restrict to rank 4 SSYT. This leaves us with a collection of $3 \times 4$ arrays whose entries increase weakly across rows and strictly down columns.

To give two examples, the SSYT in Figure 7 are valid (left) and invalid (right). Note that both are genuine SSYT of shape $3 \times 4$ with entries from $\{1, \ldots, 12\}$.

| 1 | 2 | 4 | 7 |
|---|---|---|---|
| 5 | 6 | 6 | 11 |
| 9 | 9 | 12 | 12 |

| 1 | 1 | 5 | 7 |
|---|---|---|---|
| 4 | 5 | 7 | 9 |
| 8 | 8 | 10 | 11 |

*Figure 7.* An example of a valid **(left)** and invalid **(right)** tableau from the Grassmannian cluster algebra dataset.

The dataset consists of a collection of rectangular SSYT each with a label indicating whether it indexes a cluster variable or not. Those that do not index a cluster variable are labeled with a '0' and those that do are labeled with a '1'. The valid examples are drawn from (Cheung et al., 2023) and can be obtained from https://github.com/edhirst/GrassmanniansML/. We generated our own negative examples because we found that the model learned some spurious correlations as a result of the sampling strategy used in (Cheung et al., 2023). To sample random rectangular SSYT, we took advantage of the `random_element` method in the 'Tableaux' class in Sage.

The datasets, which can be found on the GitHub page, are contained in files named:

- `3_4_12_invalid_train.txt`
- `3_4_12_invalid_test.txt`
- `3_4_12_valid_test.txt`
- `3_4_12_valid_train.txt`

In the files we use braces [ and ] to demarcate rows of the diagram, so that

`[[1, 2, 4, 7], [5, 6, 6, 11], [9, 9, 12, 12]]`

corresponds to the tableau in Figure 7, left. Dataset statistics can be found in Table 10.

## B.4. The Coefficients of Kazhdan-Lusztig Polynomials

Kazhdan-Lusztig polynomials are integer polynomials in a variable $q$ which are parameterized by two permutations. The tasks associated with this dataset are to predict the different coefficients of the polynomial (organized by monomial degree). Thus the input is two permutations $v, w$ and the output is a sequence of integers which are the coefficients of the polynomial. For instance, if $v = 0\ 2\ 1\ 3\ 5\ 4\ 6\ 9\ 7\ 8$ and $w = 2\ 3\ 0\ 5\ 9\ 6\ 7\ 8\ 1\ 4$ then

$$P_{v,w}(q) = 4q^4 + 12q^3 + 13q^2 + 6q + 1.$$

| Coefficient | Train | Test |
|---|---|---|
| 0 | 8, 496 | 2, 123 |
| 1 | 3, 024 | 757 |

*Table 11.* Statistics for the constant term in the $n = 5$ KL polynomial dataset.

| Coefficient | Train | Test |
|---|---|---|
| 0 | 11, 219 | 2, 793 |
| 1 | 267 | 77 |
| 2 | 34 | 10 |

*Table 12.* Statistics for the coefficient on $q$ in the $n = 5$ KL polynomial dataset.

and this is written as the line

```
0213546978 2305967814 1,6,13,12,4.
```

Datasets were generated using C code from Greg Warrington's website (Warrington), computing $P_{v,w}(q)$ for all pairs of permutations of size $n = 5, 6, 7$.

The files we provide are:

- `kl-polynomials_5_train.txt`

- `kl-polynomials_5_test.txt`

- `kl-polynomials_6_train.txt`

- `kl-polynomials_6_test.txt`

- `kl-polynomials_7_train.txt`

- `kl-polynomials_7_test.txt`

Statistics can be found in Table 11-Table 22.

### B.5. The Robinson-Schensted-Knuth Correspondence

The Robinson–Schensted-Knuth correspondence is a bijection between permutations and pairs of standard Young tableaux. For our initial baselines we use Young tableau pairs as input and the corresponding permutation as output, but the dataset could be used in the opposite direction.

The permutations are stored in files starting with `input_permutation` and the pair of tableaux are stored in files labeled by `output_tableau`. Permutations are stored using their inversion vectors (a binary sequence). Tableau rows are separated by '[' and ']'. For instance,

```
[[[1, 3, 4], [2, 7], [5], [6]], [[1, 2, 6], [3, 4], [5], [7]]]
```

corresponds to the pair of Young tableau in Figure 8. In our baselines, this task is formulated as regression with the aim of regressing the sequence of 0-1 entries in the inversion vector. We found that the inversion vector was easiest to work with relative to other representations of permutations in this setting.

| Coefficient | Train | Test |
|---|---|---|
| 0 | 11, 514 | 2, 876 |
| 1 | 6 | 4 |

*Table 13.* Statistics for the coefficient on $q^2$ in the $n = 5$ KL polynomial dataset.

| Coefficient | Train | Test |
|---|---|---|
| 0 | 336, 071 | 83, 922 |
| 1 | 78, 649 | 19, 758 |

*Table 14.* Statistics for the constant term in the $n = 6$ KL polynomial dataset.

| Coefficient | Train | Test |
|---|---|---|
| 0 | 397, 386 | 99, 354 |
| 1 | 13, 253 | 3, 311 |
| 2 | 3, 483 | 883 |
| 3 | 535 | 117 |
| 4 | 63 | 15 |

*Table 15.* Statistics for the coefficient on $q$ in the $n = 6$ KL polynomial dataset.

| Coefficient | Train | Test |
|---|---|---|
| 0 | 412, 707 | 103, 177 |
| 1 | 1, 705 | 441 |
| 2 | 242 | 46 |
| 3 | 40 | 8 |
| 4 | 26 | 8 |

*Table 16.* Statistics for the coefficient on $q^2$ in the $n = 6$ KL polynomial dataset.

| Coefficient | Train | Test |
|---|---|---|
| 0 | 414, 688 | 103, 670 |
| 1 | 32 | 10 |

*Table 17.* Statistics for the coefficient on $q^3$ in the $n = 6$ KL polynomial dataset.

| Coefficient | Train | Test |
|---|---|---|
| 0 | 17, 479, 910 | 4, 370, 771 |
| 1 | 2, 841, 370 | 709, 549 |

*Table 18.* Statistics for the constant term in the $n = 7$ KL polynomial dataset.

| Coefficient | Train | Test |
|---|---|---|
| 0 | 19, 291, 150 | 4, 822, 214 |
| 1 | 660, 600 | 165, 768 |
| 2 | 266, 591 | 66, 593 |
| 3 | 80, 173 | 19, 963 |
| 4 | 18, 834 | 4, 762 |
| 5 | 3, 221 | 819 |
| 6 | 711 | 201 |

*Table 19.* Statistics for the coefficient on $q$ in the $n = 7$ KL polynomial dataset.

| Coefficient | Train | Test |
|---|---|---|
| 0 | $20,072,738$ | $5,017,962$ |
| 1 | $170,412$ | $42,748$ |
| 2 | $46,226$ | $11,568$ |
| 3 | $16,227$ | $4,021$ |
| 4 | $7,621$ | $1,905$ |
| 5 | $4,023$ | $1,065$ |
| 6 | $1,287$ | $349$ |
| 7 | $1,153$ | $287$ |
| 8 | $785$ | $183$ |
| 9 | $350$ | $86$ |
| 10 | $152$ | $40$ |
| 11 | $139$ | $37$ |
| 12 | $121$ | $47$ |
| 13 | $42$ | $22$ |
| 14 | $4$ | $1$ |

*Table 20.* Statistics for the coefficient on $q^2$ in the $n = 7$ KL polynomial dataset.

| Coefficient | Train | Test |
|---|---|---|
| 0 | $20,291,535$ | $507,2831$ |
| 1 | $22,094$ | $5,498$ |
| 2 | $4,779$ | $1,213$ |
| 3 | $1,660$ | $442$ |
| 4 | $590$ | $146$ |
| 5 | $195$ | $61$ |
| 6 | $206$ | $50$ |
| 7 | $115$ | $37$ |
| 8 | $34$ | $14$ |
| 9 | $26$ | $6$ |
| 10 | $24$ | $8$ |
| 11 | $18$ | $14$ |
| 15 | $4$ | $1$ |

*Table 21.* Statistics for the coefficient on $q^3$ in the $n = 7$ KL polynomial dataset.

| Coefficient | Train | Test |
|---|---|---|
| 0 | $17,479,910$ | $4,370,771$ |
| 1 | $2,841,370$ | $709,549$ |

*Table 22.* Statistics for the coefficient on $q^4$ in the $n = 7$ KL polynomial dataset.

*Figure 8.* The Young tableau pair that are obtained by applying the Robinson-Schensted-Knuth algorithm to the permutation 6 7 2 5 3 4 1.

We store files for $n = 8, 9, 10$. These are named:

- `input_permutations_8_train.csv`

- `input_permutations_8_test.csv`

- `input_permutations_9_train.csv`

- `input_permutations_9_test.csv`

- `input_permutations_10_train.csv`

- `input_permutations_10_test.csv`

- `output_tableau_8_train.csv`

- `output_tableau_8_test.csv`

- `output_tableau_9_train.csv`

- `output_tableau_9_test.csv`

- `output_tableau_10_train.csv`

- `output_tableau_10_test.csv`

Sage (Stein et al., 2024) was used to generate the tableau pairs corresponding to each permutation. The script is available on the GitHub page.

### B.6. Schubert Polynomial Structure Constants

This task involves predicting the structure constants of Schubert polynomials. These polynomials are each indexed by permutations so structure constants are indexed by a triple of permutations. Hence, the input to the model is a triple of permutations $v, w, u$ and the output is an integer. For instance, since we have the relationship

$$\mathcal{S}_{12354}\mathcal{S}_{12354} = \mathcal{S}_{123645} + \mathcal{S}_{12453}$$

one data instance is

`[1,2,3,5,4],[1,2,3,5,4],[1,2,3,6,4,5];1.`

We partition the datasets so that the dataset associated with value $n$ has structure constants for pairs of $\mathcal{S}_v$ with $v \in S_n$. Note that there is some repetition since $S_{n-1}$ is a subset of $S_n$. We store files for $n = 4, 5, 6$. These are named:

- `schubert_structure_coefficients_triples_4_train.txt`

- `schubert_structure_coefficients_triples_4_test.txt`

|  | 0 | 1 |
|---|---|---|
| Training set | 851 | 833 |
| Testing set | 201 | 220 |

*Table 23.* Statistics for the Schubert polynomial structure constants dataset for $n = 4$.

|  | 0 | 1 | 2 |
|---|---|---|---|
| Training set | $42,831$ | $42,619$ | 170 |
| Testing set | $10,681$ | $10,680$ | 44 |

*Table 24.* Statistics for the Schubert polynomial structure constants dataset for $n = 5$.

- `schubert_structure_coefficients_triples_5_train.txt`

- `schubert_structure_coefficients_triples_5_test.txt`

- `schubert_structure_coefficients_triples_6_train.txt`

- `schubert_structure_coefficients_triples_6_test.txt.`

Sage (Stein et al., 2024) was used to generate and multiply Schubert polynomials for each pair of permutations $\alpha$ and $\beta$ in $S_n$. The basis expansion of $\mathcal{S}_\alpha \star \mathcal{S}_\beta$ was obtained from this and each nonzero term in this expansion was used as an instance. For any $n$, most structure constants will be zero. To generate a balanced dataset, we computed $\mathcal{S}_\alpha \star \mathcal{S}_\beta$ for all elements in $S_n \times S_n$ and for each $c_{\alpha,\beta}^\gamma \neq 0$, we applied a random number of transpositions (where the number of transpositions was sampled from a geometric distribution) to $\gamma$ to get $\gamma'$, checked that $c_{\alpha,\beta}^{\gamma'} = 0$ and added this to the dataset. Therefore, the dataset contains all non-zero structure constants but only a fraction of zero structure constants.

Dataset statistics can be found in Table 23-Table 25.

## B.7. Partial Orders on Lattice Paths

This dataset contains pairs of lattice paths starting at $(0,0)$ and ending at $(n, n-1)$ that are only allowed to take one unit steps either north or east, and must stay below the line $y = \frac{n}{n-1}x$. They are thus encoded by a sequence of 1's (for steps east) and 0's (for steps north) of length $(n+1) + n = 2n + 1$. Each pair of lattice paths is a covering pair in exactly one of the two partial orders, the Lagrange order or the matching order (pairs that are covers in both are few and were removed). The task is to predict which partial order a covering pair belongs to.

Each line in a file is the concatenation of two 0-1 sequences (one for each path) for a length $4n + 2$ row of 0's and 1's. The lattice paths are separated by ';'.

For an $3 \times 2$ grid, the sequence:

$$1, 1, 1, 0, 0; 1, 1, 0, 1, 0$$

corresponds to the two lattice paths in Figure 9. The first is in red and second is in blue, with segments traversed by both paths colored red.

We store files for $n = 10, 11, 12, 13$. These are named:

- `lagrange_covers_test_10_9.csv`

|  | 0 | 1 | 2 | 3 | 4 | 5 |
|---|---|---|---|---|---|---|
| Training set | $4,202,040$ | $4,093,033$ | $109,217$ | $2,262$ | 9 | 9 |
| Testing set | $1,052,062$ | $1,021,898$ | $27,110$ | 568 | 3 | 0 |

*Table 25.* Statistics for the Schubert polynomial structure constants dataset for $n = 6$.

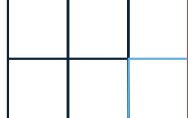

*Figure 9.* An example of two lattice paths from $(0, 0)$ to $(3, 2)$. These do not correspond to a cover relation.

|  | Lagrange order | Matching order | Total |
|---|---|---|---|
| Training set | 7, 5589 | 3, 875 | 11, 433 |
| Testing set | 1, 895 | 968 | 2, 863 |

*Table 26.* Statistics for the lattice paths dataset for paths from $(0, 0)$ to $(10, 9)$.

- `lagrange_covers_test_11_10.csv`

- `lagrange_covers_test_12_11.csv`

- `lagrange_covers_test_13_12.csv`

- `lagrange_covers_train_10_9.csv`

- `lagrange_covers_train_11_10.csv`

- `lagrange_covers_train_12_11.csv`

- `lagrange_covers_train_13_12.csv`

- `matching_covers_test_10_9.csv`

- `matching_covers_test_11_10.csv`

- `matching_covers_test_12_11.csv`

- `matching_covers_test_13_12.csv`

- `matching_covers_train_10_9.csv`

- `matching_covers_train_11_10.csv`

- `matching_covers_train_12_11.csv`

- `matching_covers_train_13_12.csv`

The first word ('Lagrange' or 'matching') gives the label, the third word says whether it is train or test, and the final two numbers give $n$ and $n - 1$.

Sage (Stein et al., 2024) was used to compute the covering pairs in the Lagrange and matching ordering; the script used to generate the data is available on the GitHub page. We removed instances that were covering pairs in both the Lagrange partial order and the matching partial order (this was 21, 40, and 79 instances for $n = 10, 11$, and 12 respectively).

The dataset statistics can be found in Table 26-Table 29.

|  | Lagrange order | Matching order | Total |
|---|---|---|---|
| Training set | 26, 427 | 13, 424 | 39, 851 |
| Testing set | 6, 601 | 3, 355 | 9, 956 |

*Table 27.* Statistics for the lattice paths dataset for paths from $(0, 0)$ to $(11, 10)$.

|              | Lagrange order | Matching order | Total    |
| ------------ | -------------- | -------------- | -------- |
| Training set | $93, 218$      | $46, 976$      | $140, 194$ |
| Testing set  | $23, 324$      | $11, 749$      | $35, 073$  |

*Table 28.* Statistics for the lattice paths dataset for paths from $(0, 0)$ to $(12, 11)$.

|              | Lagrange order | Matching order | Total    |
| ------------ | -------------- | -------------- | -------- |
| Training set | $331, 065$     | $166, 304$     | $497, 369$ |
| Testing set  | $82, 789$      | $41, 580$      | $124, 369$ |

*Table 29.* Statistics for the lattice paths dataset for paths from $(0, 0)$ to $(13, 12)$.

## B.8. Mutation Equivalence of Quivers

The task associated with this dataset is matching a quiver to one of several possible mutation equivalence classes. Thus, the input is a quiver with 11 nodes encoded by its $11 \times 11$ adjacency matrix and the label is one of 7 different equivalence classes: $A_{11}, BB_{11}, BD_{11}, BE_{11}, D_{11}, DE_{11}, E_{11}$. The files are organized by train and test for each of these classes. All mutation classes were generated using Sage (Stein et al., 2024), the code is available on the GitHub page. For the quiver mutation classes that are not mutation finite, the datasets contain quivers generated up to a certain depth, which is the distance from the original quiver, measured by number of mutations. The depth is specified in the filename and was chosen to achieve as close to a balanced dataset as possible.

The file names are:

- `A_11_bmatrices_test.csv`

- `A_11_bmatrices_train.csv`

- `BB_11_depth10_bmatrices_test.csv`

- `BB_11_depth10_bmatrices_train.csv`

- `BD_11_depth9_bmatrices_test.csv`

- `BD_11_depth9_bmatrices_train.csv`

- `BE_11_depth8_bmatrices_test.csv`

- `BE_11_depth8_bmatrices_train.csv`

- `D_11_bmatrices_test.csv`

- `D_11_bmatrices_train.csv`

- `DE_11_depth9_bmatrices_test.csv`

- `DE_11_depth9_bmatrices_train.csv`

- `E_11_depth9_bmatrices_test.csv`

- `E_11_depth9_bmatrices_train.csv`

Within a file, each row is a flattened adjacency matrix encoded in row major order. The statistics of the dataset can be found in Table 30.

|  | $A_{11}$ | $B_{11}$ | $BD_{11}$ | $BE_{11}$ | $D_{11}$ | $DE_{11}$ | $E_{11}$ | Total |
|---|---|---|---|---|---|---|---|---|
| Training set | 11,940 | 27,410 | 23,651 | 22,615 | 25,653 | 23,528 | 28,998 | 163,795 |
| Testing set | 2,984 | 6,852 | 5,912 | 5,653 | 6,413 | 5,881 | 7,249 | 409,44 |

*Table 30.* Statistics of the quiver mutation equivalence dataset.

|  | Weaving pattern | Non-weaving pattern |
|---|---|---|
| Training set | 634 | 1,116 |
| Testing set | 275 | 467 |

*Table 31.* Statistics of the weaving pattern dataset for $n = 6$.

## B.9. Weaving Patterns

Weaving patterns of size $n \times n - 1$ are a special type of matrix containing entries in $\{1, 2, \ldots, n\}$. They correspond to representations of the longest word permutation of $n$ elements (the permutation that sends $1 \mapsto n$, $2 \mapsto n - 1$, etc.). This task involves trying to identify weaving patterns among matrices that look like weaving patterns but are not.

Each matrix is stored on a single line in row-major format. For instance,

```
(0, 1, 2, 3, 3, 2, 3, 4, 2, 3, 2, 1, 5, 4, 3, 2)
```

We provide files for $n = 6, 7$. The files are:

- `weaving_patterns_6_train.txt`

- `weaving_patterns_7_train.txt`

- `weaving_patterns_6_test.txt`

- `weaving_patterns_7_test.txt`

Positive examples were generated by a program written in Java script. Negative examples were generated by loading positive examples and Python and perturbing them. Code for both of these can be found on the GitHub page.

Dataset statistics can be found in Table 31 and Table 32.

## B.10. Small Model Hyperparameters

For our baselines, we train encoder-only transformer models, standard feedforward multi-layer perceptron (MLP) models with ReLU non-linearities, and logistic regression on the classification tasks and the same architectures of transformers and MLPs along with linear regression for the regression tasks.

- To optimize MLP models we performed a simple grid search across

  ◇ learning rates: 0.001, 0.0005, and 0.0001,
  ◇ depth: 1, 2, 3, and 4,
  ◇ uniform hidden dimension: 32, 64, 128, and 256.

- To optimize the transformer models we performed a similar grid search but with hyperparameters

  ◇ learning rates: 0.001, 0.0005, and 0.0001,

|  | Weaving pattern | Non-weaving pattern |
|---|---|---|
| Training set | 17,388 | 96,012 |
| Testing set | 7,310 | 41,290 |

*Table 32.* Statistics of the weaving pattern dataset for $n = 7$.

| Dataset | Learning rate | Depth | Hidden dimension |
|---|---|---|---|
| Lattice paths | | | |
| $n = 10$ | 0.001 | 3 | 256 |
| $n = 11$ | 0.001 | 3 | 256 |
| $n = 12$ | 0.001 | 4 | 256 |
| Weaving patterns | | | |
| $n = 6$ | 0.001 | 2 | 256 |
| $n = 7$ | 0.001 | 2 | 256 |
| Cluster algebra quivers | 0.001 | 4 | 256 |
| Grassmanian cluster algebras | | | |
| $n = 6$ | 0.001 | 4 | 256 |
| Schubert polynomials | | | |
| $n = 4$ | 0.0005 | 4 | 64 |
| $n = 5$ | 0.001 | 4 | 256 |
| $n = 6$ | 0.001 | 3 | 256 |
| mHeight | | | |
| $n = 8$ | 0.0005 | 2 | 128 |
| $n = 9$ | 0.001 | 2 | 128 |
| $n = 10$ | 0.001 | 2 | 256 |
| $S_n$ characters | | | |
| $n = 18$ | 0.001 | 4 | 128 |
| $n = 20$ | 0.0001 | 4 | 256 |
| $n = 22$ | 0.001 | 4 | 256 |
| RSK | | | |
| $n = 8$ | 0.001 | 4 | 256 |
| $n = 9$ | 0.001 | 4 | 256 |

*Table 33.* Optimized hyperparameters for the MLP baselines found in Table 1 and Appendix B.10.

⋄ model dimensionality: 20, 40, and 80,

⋄ depths: 2 and 4,

⋄ number of heads: 4, 6, and 8.

MLPs and transformers were all trained for 60 epochs and the best test score was recorded. The best MLP hyperparameters for each dataset can be found in Table 33. The best transformer hyperparameters for each dataset can be found in Table 34.

All optimization was performed in Pytorch (Paszke et al., 2017) with the Adam optimizer (with Pytorch default settings beyond the learning rate) on one Nvidia $A100$. After finding an optimal hyperparameter setting, we trained 3 models, averaged their performance, and generated $95\%$ confidence intervals to measure the variability in performance with these hyperparameters.

Linear and logistic regression was performed with sklearn (Pedregosa et al., 2011) using default hyperparameters.

### B.11. LLM Evaluation Procedure

We evaluated Claude 3.5 Sonnet, GPT-4o-mini, GPT-4o, and o1-mini on the mHeight and Schubert polynomials datasets using a simple in-context learning setup and a simple program synthesis setup. When using in-context learning, we tried using 0, 10, 50, and 100 examples and also tested using chain-of-thought prompting. Finally, we explored providing the model with some background on the problem. The best setting depended on the model and task. An example template for the in-context learning prompt when chain-of-thought is used is:

```
"You are tasked with solving a classification problem.
Here is high-level information about the dataset:\n{dataset_info}\n\n"
+ f"{few_shot_str}" + "Before answering with your Python code,
reason in a step-by-step manner as to get the right answer.\n\n"
```

| Dataset | Learning rate | Depth | Hidden dimension | Heads |
|---|---|---|---|---|
| Lattice paths | | | | |
| $n = 10$ | 0.001 | 4 | 80 | 8 |
| $n = 11$ | 0.0005 | 4 | 80 | 4 |
| $n = 12$ | 0.0005 | 6 | 40 | 8 |
| Weaving patterns | | | | |
| $n = 6$ | 0.0001 | 4 | 80 | 8 |
| $n = 7$ | 0.0001 | 4 | 80 | 8 |
| Cluster algebra quivers | 0.0005 | 6 | 80 | 4 |
| Grassmanian cluster algebras | | | | |
| $n = 6$ | 0.0005 | 6 | 80 | 4 |
| Schubert polynomials | | | | |
| $n = 4$ | 0.0001 | 4 | 80 | 6 |
| $n = 5$ | 0.0005 | 4 | 40 | 6 |
| $n = 6$ | 0.0005 | 4 | 80 | 4 |
| mHeight | | | | |
| $n = 8$ | 0.0001 | 4 | 80 | 4 |
| $n = 9$ | 0.001 | 6 | 20 | 8 |
| $n = 10$ | 0.001 | 4 | 80 | 8 |
| $S_n$ characters | | | | |
| $n = 18$ | 0.001 | 6 | 80 | 6 |
| $n = 20$ | 0.001 | 4 | 80 | 8 |
| $n = 22$ | 4 | 40 | 8 | |
| RSK | | | | |
| $n = 8$ | 0.001 | 4 | 20 | 6 |
| $n = 9$ | 0.001 | 4 | 80 | 4 |

*Table 34.* Optimized hyperparameters for the transformer baselines found in Table 1 and Appendix B.10.

where 'dataset_info' is a description of the dataset and task and 'few_shot_str' are the few-shot examples. When chain-of-thought reasoning is not used the last component is changed to

```
"Do not provide any additional reasoning or explanation.
Just provide your answer at the end on its own line in the form 'ANSWER:
$ANSWER' (without quotes) where $ANSWER is the answer to the question."
```

In the code synthesis version of the experiments, models were asked to write a Python program that solves the task using only Sage (Stein et al., 2024), Numpy, and SymPy. No model ended up using either Sage or SymPy. The same prompt was used to generate 100 examples. The best program was chosen by evaluating each on the test set.

```
Your job is to write a Python function that solves the classification problem.
You will be given some examples of a classification problem from the
'{dataset}' dataset.

Write a function 'predict' that takes an input in a Python list and returns
an integer  as the classification result.

Here is information about the dataset:
{dataset_info}

Avoid using machine learning or model calls; rather, embed the logic in
Python code.
Rather than use shallow pattern matching or using simple patterns, try
to analyze the underlying combinatorial logic of the examples. Note
that the datagenerating process for this dataset is a combinatorial algorithm.
You may want to use numpy and sympy for math operations or sage for
cominatorics, however this is optional. If you do use them, *make sure
to import them within your function*.

Below are a few examples from the training set:
{training_examples}

{instructions}
Your final answer should be valid Python code enclosed in triple
backticks. This program will be evaluated on the test set.
```

All of our experiments with LLMs used AI Inspect (AI Safety Institute).

## C. Datasheets

We provide datasheet information that holds for (almost) all datasets in the collection here and provide answers that vary across each dataset in their relevant section.

- **Who funded the creation of the datasets?** This research was supported by the Mathematics for Artificial Reasoning in Science (MARS) initiative at Pacific Northwest National Laboratory. It was conducted under the Laboratory Directed Research and Development (LDRD) Program at Pacific Northwest National Laboratory (PNNL), a multiprogram National Laboratory operated by Battelle Memorial Institute for the U.S. Department of Energy under Contract DE-AC05-76RL01830.

- **Are there recommended data splits (e.g., training, development/validation, testing)?** All datasets are provided in preset splits.

- **Are there any errors, sources of noise, or redundancies in the dataset?** Not that the creators are aware of.

- **Is the dataset self-contained, or does it link to or otherwise rely on external resources (e.g., websites, tweets, other datasets)?** It is self-contained.

- **Does the dataset contain data that might be considered confidential (e.g., data that is protected by legal privilege or by doctor– patient confidentiality, data that includes the content of individuals' non-public communications)?** No.

- **Does the dataset contain data that, if viewed directly, might be offensive, insulting, threatening, or might otherwise cause anxiety?** No.

- **Over what timeframe was the data collected?** All datasets were generated between June 2024 and December 2024.

- **Were any ethical review processes conducted (e.g., by an institutional review board)?** N/A

- **Was the "raw" data saved in addition to the preprocessed/cleaned/labeled data (e.g., to support unanticipated future uses)?** No, but in most cases we supply the code to re-generate the raw data (see https://github.com/pnnl/ML4AlgComb).

- **Is the software that was used to preprocess/clean/label the data available?** Yes, https://github.com/pnnl/ML4AlgComb.

- **Is there a repository that links to any or all papers or systems that use the dataset?** Yes, https://github.com/pnnl/ML4AlgComb.

- **How will the dataset will be distributed (e.g., tarball on website, API, GitHub)? Does the dataset have a digital object identifier (DOI)?** All datasets will be compressed to a single .zip file and stored on Google Drive. It does not currently have a DOI.

## C.1. Symmetric group characters datasheet

- **For what purpose was the dataset created?** To study whether machine learning models can learn to predict the characters of irreducible representations of the symmetric group.

- **Who created the dataset (e.g., which team, research group) and on behalf of which entity (e.g., company, institution, organization)?** The dataset was created by Henry Kvinge at Pacific Northwest National Laboratory.

- **What do the instances that comprise the dataset represent (e.g., documents, photos, people, countries)?** Characters of irreducible representations of the symmetric group of size $n$.

- **How many instances are there in total (of each type, if appropriate)?** See Appendix B.1.

- **Does the dataset contain all possible instances or is it a sample (not necessarily random) of instances from a larger set?** It contains all possible instances for a fixed $n$.

- **What data does each instance consist of?** Each instance consists of two partitions of $n$ and an integer.

- **Is there a label or target associated with each instance?** Yes, the label is the final integer.

- **Is any information missing from individual instances?** No.

- **Are relationships between individual instances made explicit (e.g., users' movie ratings, social network links)?** N/A

- **How was the data associated with each instance acquired?** The creators used the open-source mathematics software system SageMath to calculate character values.

- **What mechanisms or procedures were used to collect the data (e.g., hardware apparatuses or sensors, manual human curation, software programs, software APIs)?** SageMath was used for the calculations and Python was used to sort, format, split the data. This was all done on a consumer laptop.

- **If the dataset is a sample from a larger set, what was the sampling strategy (e.g., deterministic, probabilistic with specific sampling probabilities)?** N/A

- **Who was involved in the data collection process (e.g., students, crowdworkers, contractors) and how were they compensated (e.g., how much were crowdworkers paid)?** Henry Kvinge wrote the code in SageMath to generate this dataset.

- **Was any preprocessing/cleaning/labeling of the data done (e.g., discretization or bucketing, tokenization, part-of-speech tagging, SIFT feature extraction, removal of instances, processing of missing values)?** Partitions are listed as sequences of numbers enclosed by brackets (see Appendix B.1.

- **Has the dataset been used for any tasks already?** No.

- **What (other) tasks could the dataset be used for?** This dataset could be used in instances where the aim is to understand the ability of machine learning to perform challenging mathematical tasks.

- **Is there anything about the composition of the dataset or the way it was collected and preprocessed/cleaned/labeled that might impact future uses?** No.

- **Are there tasks for which the dataset should not be used? If so, please provide a description.** No.

- **Will the dataset be distributed to third parties outside of the entity (e.g., company, institution, organization) on behalf of which the dataset was created? If so, please provide a description.** No.

- **How will the dataset will be distributed (e.g., tarball on website, API, GitHub)? Does the dataset have a digital object identifier (DOI)?** All datasets will be compressed to a single `.zip` file and stored on Google Drive. It does not currently have a DOI.

- **Will the dataset be distributed under a copyright or other intellectual property (IP) license, and/or under applicable terms of use (ToU)?** CC0, https://creativecommons.org/public-domain/cc0/

- **Have any third parties imposed IP-based or other restrictions on the data associated with the instances?** No.

## C.2. mHeight datasheet

- **For what purpose was the dataset created?** This dataset was created to study a machine learning model's ability to learn the mHeight of a permutation.

- **Who created the dataset (e.g., which team, research group) and on behalf of which entity (e.g., company, institution, organization)?** The dataset was created by Herman Chau at the University of Washington.

- **What do the instances that comprise the dataset represent (e.g., documents, photos, people, countries)?** Instances are permutations (represented in one-line notation) followed by the integer corresponding to their mHeight.

- **How many instances are there in total (of each type, if appropriate)?** See Appendix B.2.

- **Does the dataset contain all possible instances or is it a sample (not necessarily random) of instances from a larger set?** Yes, all instances are included for the values of $n$ provided.

- **What data does each instance consist of?** An instance consists of a permutation and its corresponding mHeight.

- **Is there a label or target associated with each instance?** The label is the mHeight.

- **Is any information missing from individual instances?** No.

- **Are relationships between individual instances made explicit (e.g., users' movie ratings, social network links)?** Individual instances are unrelated.

- **How was the data associated with each instance acquired?** The creators generated the data using Python scripts which are provided at https://github.com/pnnl/ML4AlgComb.

- **What mechanisms or procedures were used to collect the data (e.g., hardware apparatuses or sensors, manual human curation, software programs, software APIs)?** Python was used to generate the data on a consumer laptop.

- **If the dataset is a sample from a larger set, what was the sampling strategy (e.g., deterministic, probabilistic with specific sampling probabilities)?** N/A

- **Who was involved in the data collection process (e.g., students, crowdworkers, contractors) and how were they compensated (e.g., how much were crowdworkers paid)?** Herman Chau wrote Python code to generate the data.

- **Was any preprocessing/cleaning/labeling of the data done (e.g., discretization or bucketing, tokenization, part-of-speech tagging, SIFT feature extraction, removal of instances, processing of missing values)?** No.

- **Has the dataset been used for any tasks already?** No.

- **What (other) tasks could the dataset be used for?** This dataset could be used to train a model to predict the mHeight of permutations and as an intermediate task for predicting the smallest non-trivial zero coefficient of Kazhdan-Lusztig polynomials.

- **Is there anything about the composition of the dataset or the way it was collected and preprocessed/cleaned/labeled that might impact future uses?** No.

- **Are there tasks for which the dataset should not be used? If so, please provide a description.** No.

- **Will the dataset be distributed to third parties outside of the entity (e.g., company, institution, organization) on behalf of which the dataset was created? If so, please provide a description.** No.

- **How will the dataset will be distributed (e.g., tarball on website, API, GitHub)? Does the dataset have a digital object identifier (DOI)?** All datasets will be compressed to a single `.zip` file and stored on Google Drive. It does not currently have a DOI.

- **Will the dataset be distributed under a copyright or other intellectual property (IP) license, and/or under applicable terms of use (ToU)?** CC0, `https://creativecommons.org/public-domain/cc0/`

- **Have any third parties imposed IP-based or other restrictions on the data associated with the instances?** No.

### C.2.1. Semistandard Young tableau for Grassmannian datasheet

- **For what purpose was the dataset created?** This dataset was created to study a machine learning model's ability to identify whether an semistandard Young tableau indexes a valid cluster variable in the Grassmannian cluster algebra.

- **Who created the dataset (e.g., which team, research group) and on behalf of which entity (e.g., company, institution, organization)?** The code for the positive examples in the dataset was created by an external team consisting of Man-Wai Cheung, Pierre-Phillips Dechant, Yang-Hui He, Elli Heyes, Edward Hirst, and Jian-Rong Li (Cheung et al., 2023). The negative dataset was created by Herman Chau at the University of Washington.

- **What do the instances that comprise the dataset represent (e.g., documents, photos, people, countries)?** Instances represent semistandard young tableau.

- **How many instances are there in total (of each type, if appropriate)?** See Appendix B.3.

- **Does the dataset contain all possible instances or is it a sample (not necessarily random) of instances from a larger set?** The positive instances are all possible instances with exceedingly high probability. An equal number of negative instances were obtained by randomly sampling without duplicates. See Appendix B.3 for details on sampling.

- **What data does each instance consist of?** An instance consists of a $3 \times 4$ SSYT.

- **Is there a label or target associated with each instance?** The label is given in the filename.

- **Is any information missing from individual instances?** No.

- **Are relationships between individual instances made explicit (e.g., users' movie ratings, social network links)?** Individual instances are unrelated.

- **How was the data associated with each instance acquired?** The creators generated the data using Python scripts.

- **What mechanisms or procedures were used to collect the data (e.g., hardware apparatuses or sensors, manual human curation, software programs, software APIs)?** Python was used to generate the data.

- **If the dataset is a sample from a larger set, what was the sampling strategy (e.g., deterministic, probabilistic with specific sampling probabilities)?** Positive examples are generated probabilistically until no new instances are generated after a large number of iterations. Negative examples are sampled uniformly at random. See Appendix B.3 for details on sampling.

- **Who was involved in the data collection process (e.g., students, crowdworkers, contractors) and how were they compensated (e.g., how much were crowdworkers paid)?** Herman Chau generated this data. He developed his own code to generate the negative examples and used code from Man-Wai Cheung, Pierre-Phillips Dechant, Yang-Hui He, Elli Heyes, Edward Hirst, and Jian-Rong Li (Cheung et al., 2023) to generate positive examples.

- **Was any preprocessing/cleaning/labeling of the data done (e.g., discretization or bucketing, tokenization, part-of-speech tagging, SIFT feature extraction, removal of instances, processing of missing values)?** Semistandard Young tableau were flattened to fit on a single line in the data files.

- **Has the dataset been used for any tasks already?** A similar dataset was used in (Cheung et al., 2023), with different negative examples.

- **What (other) tasks could the dataset be used for?** This dataset could be used for any tasks around the study of the Grassmannian cluster algebra.

- **Is there anything about the composition of the dataset or the way it was collected and preprocessed/cleaned/labeled that might impact future uses?** No.

- **Are there tasks for which the dataset should not be used? If so, please provide a description.** No.

- **Will the dataset be distributed to third parties outside of the entity (e.g., company, institution, organization) on behalf of which the dataset was created? If so, please provide a description.** No.

- **Will the dataset be distributed under a copyright or other intellectual property (IP) license, and/or under applicable terms of use (ToU)?** CC0, `https://creativecommons.org/public-domain/cc0/`

- **Have any third parties imposed IP-based or other restrictions on the data associated with the instances?** No.

### C.2.2. KAZHDAN-LUSZTIG POLYNOMIAL DATASHEET

- **For what purpose was the dataset created?** This dataset was created to study a machine learning model's ability to learn and predict the coefficients of Kazhdan-Lusztig polynomials.

- **Who created the dataset (e.g., which team, research group) and on behalf of which entity (e.g., company, institution, organization)?** Henry Kvinge created these datasets using code from (Warrington).

- **What do the instances that comprise the dataset represent (e.g., documents, photos, people, countries)?** An instance consists of a pair of permutations followed by the Kazhdan-Lusztig polynomial corresponding to the pair of permutations. The polynomial is listed via its coefficients (starting with the constant term) up to the largest non-zero coefficient.

- **How many instances are there in total (of each type, if appropriate)?** See section Appendix B.4.

- **Does the dataset contain all possible instances or is it a sample (not necessarily random) of instances from a larger set?** For a given $n$, all KL polynomials are included.

- **What data does each instance consist of?** An instance consists of a pair of permutations followed by the Kazhdan-Lusztig polynomial corresponding to the pair of permutations.

- **Is there a label or target associated with each instance?** The labels can be taken to be the coefficients, the degree, etc.

- **Is any information missing from individual instances?** No.

- **Are relationships between individual instances made explicit (e.g., users' movie ratings, social network links)?** Individual instances are unrelated.

- **How was the data associated with each instance acquired?** The creators generated the data using C code from (Warrington) and Python scripts.

- **What mechanisms or procedures were used to collect the data (e.g., hardware apparatuses or sensors, manual human curation, software programs, software APIs)?** C code from (Warrington) and Python was used to generate the data.

- **If the dataset is a sample from a larger set, what was the sampling strategy (e.g., deterministic, probabilistic with specific sampling probabilities)?** N/A.

- **Who was involved in the data collection process (e.g., students, crowdworkers, contractors) and how were they compensated (e.g., how much were crowdworkers paid)?** Henry Kvinge generated the data.

- **Was any preprocessing/cleaning/labeling of the data done (e.g., discretization or bucketing, tokenization, part-of-speech tagging, SIFT feature extraction, removal of instances, processing of missing values)?** Polynomials are stored as a sequence of coefficients ending with the final non-zero coefficient.

- **Has the dataset been used for any tasks already?** No.

- **What (other) tasks could the dataset be used for?** This dataset could be used to study various properties of the coefficients of Kazhdan-Lusztig polynomials.

- **Is there anything about the composition of the dataset or the way it was collected and preprocessed/cleaned/labeled that might impact future uses?** No.

- **Are there tasks for which the dataset should not be used? If so, please provide a description.** No.

- **Will the dataset be distributed to third parties outside of the entity (e.g., company, institution, organization) on behalf of which the dataset was created? If so, please provide a description.** No.

- **How will the dataset will be distributed (e.g., tarball on website, API, GitHub)? Does the dataset have a digital object identifier (DOI)?** All datasets will be compressed to a single `.zip` file and stored on Google Drive. It does not currently have a DOI.

- **Will the dataset be distributed under a copyright or other intellectual property (IP) license, and/or under applicable terms of use (ToU)?** CC0, `https://creativecommons.org/public-domain/cc0/`

- **Have any third parties imposed IP-based or other restrictions on the data associated with the instances?** No.

### C.2.3. ROBINSON-SCHENSTED DATASHEET

- **For what purpose was the dataset created?** This dataset was created to study a machine learning model's ability to learn the Robinson-Schensted correspondence.

- **Who created the dataset (e.g., which team, research group) and on behalf of which entity (e.g., company, institution, organization)?** The dataset was created by Henry Kvinge and Helen Jenne at Pacific Northwest National Laboratory.

- **What do the instances that comprise the dataset represent (e.g., documents, photos, people, countries)?** Each instance in the dataset is a pair of standard Young Tableaux of the same shape, along with their associated permutation under the Robinson-Schensted-Knuth correspondence.

- **How many instances are there in total (of each type, if appropriate)?** See Appendix B.5.

- **Does the dataset contain all possible instances or is it a sample (not necessarily random) of instances from a larger set?** The dataset contains all possible instances for $n = 8, 9$, and $10$.

- **What data does each instance consist of?** An instance in the dataset consists of a pair of standard young tableaux along with its associated permutation.

- **Is there a label or target associated with each instance?** The target permutations are given in separate files.

- **Is any information missing from individual instances?** No.

- **Are relationships between individual instances made explicit (e.g., users' movie ratings, social network links)?** N/A

- **How was the data associated with each instance acquired?** The creators used the open-source mathematics software system SageMath to generate the tableau pairs corresponding to each permutation. (See `https://doc.sagemath.org/html/en/reference/combinat/sage/combinat/rsk.html` for the relevant documentation).

- **What mechanisms or procedures were used to collect the data (e.g., hardware apparatuses or sensors, manual human curation, software programs, software APIs)?** SageMath was used to generate the data and Python was used to sort, format, and split the data.

- **If the dataset is a sample from a larger set, what was the sampling strategy (e.g., deterministic, probabilistic with specific sampling probabilities)?** N/A

- **Who was involved in the data collection process (e.g., students, crowdworkers, contractors) and how were they compensated (e.g., how much were crowdworkers paid)?** Henry Kvinge and Helen Jenne used code from SageMath to generate this dataset. The author of the original implementation of the Robinson-Schensted-Knuth correspondence in SageMath QuiverMutationType class is Travis Scrimshaw.

- **Was any preprocessing/cleaning/labeling of the data done (e.g., discretization or bucketing, tokenization, part-of-speech tagging, SIFT feature extraction, removal of instances, processing of missing values)?** Standard Young tableau were flattened to fit on a single line. Permutations are written as a list of numbers.

- **Has the dataset been used for any tasks already?** The Robinson-Schensted correspondence is an important algorithm in the algebraic combinatorics community, but this specific dataset has not been used before.

- **What (other) tasks could the dataset be used for?** This dataset could be used for any tasks involving the RSK algorithm.

- **Is there anything about the composition of the dataset or the way it was collected and preprocessed/cleaned/labeled that might impact future uses?** No.

- **Are there tasks for which the dataset should not be used? If so, please provide a description.** No.

- **Will the dataset be distributed to third parties outside of the entity (e.g., company, institution, organization) on behalf of which the dataset was created? If so, please provide a description.** No.

- **How will the dataset will be distributed (e.g., tarball on website, API, GitHub)? Does the dataset have a digital object identifier (DOI)?** All datasets will be compressed to a single `.zip` file and stored on Google Drive. It does not currently have a DOI.

- **Will the dataset be distributed under a copyright or other intellectual property (IP) license, and/or under applicable terms of use (ToU)?** CC0, `https://creativecommons.org/public-domain/cc0/`

- **Have any third parties imposed IP-based or other restrictions on the data associated with the instances?** No.

### C.2.4. SCHUBERT POLYNOMIAL STRUCTURE COEFFICIENTS DATASHEET

- **For what purpose was the dataset created?** This dataset was created to study machine learning model's ability to predict the structure constants of Schubert polynomials.

- **Who created the dataset (e.g., which team, research group) and on behalf of which entity (e.g., company, institution, organization)?** The dataset was created by Henry Kvinge and Helen Jenne at Pacific Northwest National Laboratory.

- **What do the instances that comprise the dataset represent (e.g., documents, photos, people, countries)?** Instances represent the structure constants that come from multiplying Schubert polynomials together. For fixed $n$, instance

$$[\alpha, \beta, \gamma, c]$$

  where $\alpha, \beta \in S_n$, $\gamma$ is another permutation (possibly in a larger or smaller symmetric group), and $c \in \mathbb{Z}_{\geq 0}$.

- **How many instances are there in total (of each type, if appropriate)?** See Appendix B.6.

- **Does the dataset contain all possible instances or is it a sample (not necessarily random) of instances from a larger set?** For any $n$, most structure constants will be zero. To generate a balanced dataset, we computed $\mathcal{S}_\alpha \star \mathcal{S}_\beta$ for all elements in $S_n \times S_n$ and for each $c_{\alpha,\beta}^\gamma \neq 0$, we applied a random number of transpositions (where the number of transpositions was sampled from a geometric distribution) to $\gamma$ to get $\gamma'$, checked that $c_{\alpha,\beta}^{\gamma'} = 0$ and added this to the dataset. Therefore the dataset contains all non-zero structure constants but only a fraction of zero structure constants.

- **What data does each instance consist of?** An instance in the dataset corresponding to $n$ consists of two permutations from $S_n$, a permutation from another (possibly larger or smaller symmetric group), and an integer.

- **Is there a label or target associated with each instance?** The final integer in the instance is the label.

- **Is any information missing from individual instances?** No.

- **Are relationships between individual instances made explicit (e.g., users' movie ratings, social network links)?** Instances that have the same first two permutations $\alpha, \beta$ are drawn from the same basis expansion of $\mathcal{S}_\alpha \star \mathcal{S}_\beta$.

- **How was the data associated with each instance acquired?** The creators used the open-source mathematics software system SageMath to generate and multiply Schubert polynomials for each pair of permutations $\alpha$ and $\beta$ in $S_n$ for $n = 4, 5, 6$. The basis expansion of $\mathcal{S}_\alpha \star \mathcal{S}_\beta$ was obtained from this and each term in this expansion was used as an instance.

- **What mechanisms or procedures were used to collect the data (e.g., hardware apparatuses or sensors, manual human curation, software programs, software APIs)?** SageMath was used for the calculations and Python was used to sort, format, split the data.

- **If the dataset is a sample from a larger set, what was the sampling strategy (e.g., deterministic, probabilistic with specific sampling probabilities)?** For any $n$, most structure constants will be zero. To generate a balanced dataset, we computed $\mathcal{S}_\alpha \star \mathcal{S}_\beta$ for all elements in $S_n \times S_n$ and for each $c_{\alpha,\beta}^\gamma \neq 0$, we applied a random number of transpositions (where the number of transpositions was sampled from a geometric distribution) to $\gamma$ to get $\gamma'$, checked that $c_{\alpha,\beta}^{\gamma'} = 0$ and added this to the dataset. Therefore the dataset contains all non-zero structure constants but only a fraction of zero structure constants.

- **Who was involved in the data collection process (e.g., students, crowdworkers, contractors) and how were they compensated (e.g., how much were crowdworkers paid)?** Henry Kvinge and Helen Jenne wrote the code in SageMath to generate this dataset.

- **Was any preprocessing/cleaning/labeling of the data done (e.g., discretization or bucketing, tokenization, part-of-speech tagging, SIFT feature extraction, removal of instances, processing of missing values)?** Permutations are represented as lists of integers enclosed by brackets.

- **Has the dataset been used for any tasks already?** Schubert structure constants are an area of intense interest to the algebraic combinatorics community, but this specific dataset has never been used before.

- **What (other) tasks could the dataset be used for?** This dataset could be used for any tasks around the study of Schubert polynomial structure constants.

- **Is there anything about the composition of the dataset or the way it was collected and preprocessed/cleaned/labeled that might impact future uses?** No.

- **Are there tasks for which the dataset should not be used? If so, please provide a description.** No.

- **Will the dataset be distributed to third parties outside of the entity (e.g., company, institution, organization) on behalf of which the dataset was created? If so, please provide a description.** No.

- **How will the dataset will be distributed (e.g., tarball on website, API, GitHub)? Does the dataset have a digital object identifier (DOI)?** All datasets will be compressed to a single `.zip` file and stored on Google Drive. It does not currently have a DOI.

- **Will the dataset be distributed under a copyright or other intellectual property (IP) license, and/or under applicable terms of use (ToU)?** CC0, https://creativecommons.org/public-domain/cc0/

- **Have any third parties imposed IP-based or other restrictions on the data associated with the instances?** No.

C.2.5. PARTIAL ORDERS ON LATTICE PATHS DATASHEET

- **For what purpose was the dataset created?** This dataset was created to study a machine learning model's ability to differentiate the Lagrange and matching partial orders on lattice paths from $(0,0)$ to $(n, n-1)$ which do not pass above the diagonal.

- **Who created the dataset (e.g., which team, research group) and on behalf of which entity (e.g., company, institution, organization)?** The dataset was created by Helen Jenne at Pacific Northwest National Laboratory.

- **What do the instances that comprise the dataset represent (e.g., documents, photos, people, countries)?** Instances represent two lattice paths $(p, q)$, where $q$ is a cover of $p$ in either the Lagrange ordering or the matching ordering.

- **How many instances are there in total (of each type, if appropriate)?** See Appendix B.7.

- **Does the dataset contain all possible instances or is it a sample (not necessarily random) of instances from a larger set?** The dataset contains the vast majority of possible instances, but pairs $(p, q)$ that were covering pairs in both the Lagrange partial order and the matching partial order were thrown out (this was 21, 40, 79, and 183 instances for $n = 10, 11, 12$, and $13$, respectively)

- **What data does each instance consist of?** An instance in the dataset consists of two lattice paths represented as binary sequences.

- **Is there a label or target associated with each instance?** The label is given in the filename; the Lagrange and matching covers are saved in separate files.

- **Is any information missing from individual instances?** No.

- **Are relationships between individual instances made explicit (e.g., users' movie ratings, social network links)?**

  Each lattice path does not necessarily have a unique cover, so there are instances in the dataset that have the same first lattice path. This is the reason for the dataset imbalance: lattice paths have unique covers less often in the Lagrange partial ordering.

- **How was the data associated with each instance acquired?** The creators used the open-source mathematics software system SageMath to generate all lattice paths from $(0,0)$ to $(n, n-1)$ that stay below the diagonal $y = \frac{n}{n-1} x$, and compute the Lagrange number $L(p)$ and matching number $M(p)$ associated to each lattice path $p$. The matching order (resp. Lagrange order) dataset consists of lattice paths $(p, q)$ such that $M(q) > M(p)$ (resp. $L(q) > L(p)$) and there is not a path $r$ such that $M(q) > M(r) > M(p)$ (resp. $L(q) > L(r) > L(p)$).

- **What mechanisms or procedures were used to collect the data (e.g., hardware apparatuses or sensors, manual human curation, software programs, software APIs)?** SageMath was used for the calculations and Python was used to sort, format, and split the data.

- **If the dataset is a sample from a larger set, what was the sampling strategy (e.g., deterministic, probabilistic with specific sampling probabilities)?** The dataset is not a sample from a larger dataset, but we did throw out instances that are covering pairs in both the Lagrange partial order and the matching partial order (this was 21, 40, and 79 instances for $n = 10, 11$, and $12$ respectively)

- **Who was involved in the data collection process (e.g., students, crowdworkers, contractors) and how were they compensated (e.g., how much were crowdworkers paid)?** Helen Jenne wrote the code in SageMath to generate this dataset.

- **Was any preprocessing/cleaning/labeling of the data done (e.g., discretization or bucketing, tokenization, part-of-speech tagging, SIFT feature extraction, removal of instances, processing of missing values)?** We converted lattice paths to binary codes of 0's and 1's for storage.

- **Has the dataset been used for any tasks already?** The Lagrange and matching orderings are an area of recent interest in the algebraic combinatorics community, but this specific dataset has never been used before.

- **What (other) tasks could the dataset be used for?** This dataset could be used for any tasks around the study of the Lagrange and matching orderings.

- **Is there anything about the composition of the dataset or the way it was collected and preprocessed/cleaned/labeled that might impact future uses?** No.

- **Are there tasks for which the dataset should not be used? If so, please provide a description.** No.

- **Will the dataset be distributed to third parties outside of the entity (e.g., company, institution, organization) on behalf of which the dataset was created? If so, please provide a description.** No.

- **How will the dataset will be distributed (e.g., tarball on website, API, GitHub)? Does the dataset have a digital object identifier (DOI)?** All datasets will be compressed to a single `.zip` file and stored on Google Drive. It does not currently have a DOI.

- **Will the dataset be distributed under a copyright or other intellectual property (IP) license, and/or under applicable terms of use (ToU)?** CC0, `https://creativecommons.org/public-domain/cc0/`

- **Have any third parties imposed IP-based or other restrictions on the data associated with the instances?** No.

### C.2.6. MUTATION EQUIVALENT QUIVERS DATASHEET

- **For what purpose was the dataset created?** To study the problem of determining whether two quivers are mutation equivalent.

- **Who created the dataset (e.g., which team, research group) and on behalf of which entity (e.g., company, institution, organization)?** The dataset was created by Helen Jenne at Pacific Northwest National Laboratory.

- **What do the instances that comprise the dataset represent (e.g., documents, photos, people, countries)?** The instances are $11 \times 11$ adjacency matrices corresponding to 11 vertex quivers.

- **How many instances are there in total (of each type, if appropriate)?** See Appendix B.8.

- **Does the dataset contain all possible instances or is it a sample (not necessarily random) of instances from a larger set?** For the mutation classes $A$ and $D$, the dataset contains all possible instances. The other mutation classes are not finite, so it is not possible to generate all instances. For these mutation classes, we specify a depth $d$ and then generate the dataset so that it contains all quivers at most $d$ mutations away from the original quiver.

- **What data does each instance consist of?** Each instance consists of an $11 \times 11$ adjacency matrix which represents an 11 vertex quiver.

- **Is there a label or target associated with each instance?** Yes, the label is specified in the file name in which the instance is stored.

- **Is any information missing from individual instances?** No.

- **Are relationships between individual instances made explicit (e.g., users' movie ratings, social network links)?** Yes

- **How was the data associated with each instance acquired?** The creators used the open-source mathematics software system SageMath to generate all quivers from each mutation class up to the specified depth. (See https://doc.sagemath.org/html/en/reference/combinat/sage/combinat/cluster_algebra_quiver/quiver_mutation_type.html for the relevant documentation).

- **What mechanisms or procedures were used to collect the data (e.g., hardware apparatuses or sensors, manual human curation, software programs, software APIs)?** SageMath was used for the calculations and Python was used to sort, format, and split the data. Computations were done on a consumer laptop.

- **If the dataset is a sample from a larger set, what was the sampling strategy (e.g., deterministic, probabilistic with specific sampling probabilities)?** Sampling was done by applying 9 uniform, randomly sampled mutations of the quiver in non-finite type cases. For finite type all examples are included.

- **Who was involved in the data collection process (e.g., students, crowdworkers, contractors) and how were they compensated (e.g., how much were crowdworkers paid)?** Helen Jenne used code from SageMath to generate this dataset. The authors of the SageMath QuiverMutationType class are Gregg Musiker and Christian Stump, and Hugh Thomas.

- **Was any preprocessing/cleaning/labeling of the data done (e.g., discretization or bucketing, tokenization, part-of-speech tagging, SIFT feature extraction, removal of instances, processing of missing values)?** Graphs were converted to adjacency matrices and then flattened for storage.

- **Has the dataset been used for any tasks already?** A similar dataset was used in (Bao et al., 2020) to study the ability of Naive Bayes and convolutional neural networks to classify quivers according to mutation class. Their dataset also included type $A$, $D$, and $E$ mutation classes, but for smaller values of $n$ (the number of vertices) and consequently, fewer instances of each class. (Bao et al., 2020) inspired this dataset, as it seemed interesting to investigate whether better classification performance could be achieved with larger values of $n$. (He et al., 2024) used a subset of this dataset to rediscover some known characterization theorems from quiver mutation equivalence classes.

- **What (other) tasks could the dataset be used for?** This dataset could be used for any tasks focused on matching quivers to their mutation equivalence class.

- **Is there anything about the composition of the dataset or the way it was collected and preprocessed/cleaned/labeled that might impact future uses?** No.

- **Are there tasks for which the dataset should not be used? If so, please provide a description.** No.

- **Will the dataset be distributed to third parties outside of the entity (e.g., company, institution, organization) on behalf of which the dataset was created? If so, please provide a description.** No.

- **How will the dataset will be distributed (e.g., tarball on website, API, GitHub)? Does the dataset have a digital object identifier (DOI)?** All datasets will be compressed to a single .zip file and stored on Google Drive. It does not currently have a DOI.

- **Will the dataset be distributed under a copyright or other intellectual property (IP) license, and/or under applicable terms of use (ToU)?** CC0, https://creativecommons.org/public-domain/cc0/

- **Have any third parties imposed IP-based or other restrictions on the data associated with the instances?** No.

### C.2.7. WEAVING PATTERNS DATASHEET

- **For what purpose was the dataset created?** This dataset was created to study machine learning model's ability to illuminate properties of weaving patterns.

- **Who created the dataset (e.g., which team, research group) and on behalf of which entity (e.g., company, institution, organization)?** The dataset was created by Herman Chau of the University of Washington (true weaving patterns) and Davis Brown of Pacific Northwest National Laboratory (false weaving patterns).

- **What do the instances that comprise the dataset represent (e.g., documents, photos, people, countries)?** Instances consist of a $\{1, 2, \ldots, n\}$-valued matrix followed by a 0 if the matrix is a weaving pattern and a 1 if not.

- **How many instances are there in total (of each type, if appropriate)?** See Appendix B.9.

- **Does the dataset contain all possible instances or is it a sample (not necessarily random) of instances from a larger set?** Each dataset contains all weaving patterns. The non-weaving patterns are a sample from all $\{1, 2, \ldots, n\}$-valued matrices obtained by permuting two of the entries in the row of a weaving pattern and checking that the resulting matrix is not a weaving pattern.

- **What data does each instance consist of?** Instances consist of a $\{1, 2, \ldots, n\}$-valued matrix followed by a $0$ if the matrix is a weaving pattern and a $1$ if not.

- **Is there a label or target associated with each instance?** The final $0$ or $1$ in the row is the label.

- **Is any information missing from individual instances?** No.

- **Are relationships between individual instances made explicit (e.g., users' movie ratings, social network links)?** No.

- **How was the data associated with each instance acquired?** All examples were generated using a Python script.

- **What mechanisms or procedures were used to collect the data (e.g., hardware apparatuses or sensors, manual human curation, software programs, software APIs)?** All examples were generated using a Python script.

- **If the dataset is a sample from a larger set, what was the sampling strategy (e.g., deterministic, probabilistic with specific sampling probabilities)?** Each dataset contains all weaving patterns. The non-weaving patterns are a sample from all $\{1, 2, \ldots, n\}$-valued matrices obtained by permuting two of the entries in the row of a weaving pattern and checking that the resulting matrix is not a weaving pattern.

- **Who was involved in the data collection process (e.g., students, crowdworkers, contractors) and how were they compensated (e.g., how much were crowdworkers paid)?** Code was written by Herman Chau of the University of Washington (true weaving patterns) and Davis Brown of Pacific Northwest National Laboratory (false weaving patterns).

- **Was any preprocessing/cleaning/labeling of the data done (e.g., discretization or bucketing, tokenization, part-of-speech tagging, SIFT feature extraction, removal of instances, processing of missing values)?** Matrices were flattened to row-major format.

- **Has the dataset been used for any tasks already?** No.

- **What (other) tasks could the dataset be used for?** The dataset could be used for other tasks related to the study of weaving patterns.

- **Is there anything about the composition of the dataset or the way it was collected and preprocessed/cleaned/labeled that might impact future uses?** No.

- **Are there tasks for which the dataset should not be used? If so, please provide a description.** No.

- **Will the dataset be distributed to third parties outside of the entity (e.g., company, institution, organization) on behalf of which the dataset was created? If so, please provide a description.** No.

- **How will the dataset will be distributed (e.g., tarball on website, API, GitHub)? Does the dataset have a digital object identifier (DOI)?** All datasets will be compressed to a single `.zip` file and stored on Google Drive. It does not currently have a DOI.

- **Will the dataset be distributed under a copyright or other intellectual property (IP) license, and/or under applicable terms of use (ToU)?** CC0, https://creativecommons.org/public-domain/cc0/

- **Have any third parties imposed IP-based or other restrictions on the data associated with the instances?** No.

