# OpenReview forum: "Machine Learning meets Algebraic Combinatorics: A Suite of Datasets Capturing Research-level Conjecturing Ability in Pure Mathematics"
_ICML.cc/2025/Conference — ICML 2025 oral_

### Official Review · Reviewer_eme9 · 2025-02-28

**Overall Recommendation:** 4

**Summary:**

(i) The paper introduces the Algebraic Combinatorics Dataset Repository (ACD Repo), a collection of nine datasets designed to challenge machine learning models in conjecturing and problem-solving in modern algebraic combinatorial mathematics. (ii) This dataset contains foundational results and open problems in algebraic combinatorics, including topics such as computing characters of irreducible symmetric group representations, mHeight function of a permutation, and Schubert polynomial structure constants. (iii) The paper evaluates various machine learning models, including logistic regression, MLPs, and transformers, highlighting challenges in interpretability and performance.

**Claims And Evidence:**

The claims about novelty and difficulty of dataset are justified, but I think the following claims are problematic.
(i) The paper claims that ML models can generate useful conjectures in algebraic combinatorics. However, there is not explicit example of a novel conjecture validated by human experts is provided.
(ii) The effectiveness of different ML models is demonstrated, but unexpected performance gaps (e.g., transformers performing worse than MLPs in the lattice path dataset) are not sufficiently analyzed.
(iii) The claim that dataset diversity aids in generalization is plausible, but its empirical justification is limited to performance metrics rather than deeper mathematical insights.

**Essential References Not Discussed:**

It seems that the paper includes all essential references in related works.

**Experimental Designs Or Analyses:**

As shown in the Methods and Evaluation Criteria section, while results are clearly presented, deeper analysis is missing in cases where models underperform, such as the transformer results in Table 1. Furthermore, more discussion on the effect of dataset difficulty and feature representations on model success rates would enhance the findings.

**Methods And Evaluation Criteria:**

Overall, the methods and evaluation make sense. The datasets are well-structured, covering a range of combinatorial problems suitable for ML-based conjecturing. Evaluation metrics focus on accuracy, but additional measures such as model interpretability and robustness could strengthen the analysis. Some datasets are relatively small, raising concerns about generalizability to more complex mathematical settings.

**Other Comments Or Suggestions:**

Below are some suggestions for improvement:
(i) Discuss the computational cost of training models on the datasets.
(ii) Provide qualitative examples of conjectures generated by ML models that are non-trivial and potentially useful to mathematicians.
(iii) Moves the appendix pdf from supplementary material to the paper.

**Other Strengths And Weaknesses:**

Strengths: The paper contributes novel and cutting-edge datasets with real mathematical significance. The datasets are diverse and encourage broader engagement with ML in mathematics.
Weaknesses: The lack of discussion on surprising experimental results and the absence of clear examples of ML-generated conjectures that have led to mathematical insights.

**Questions For Authors:**

(i) Why do transformers perform so poorly on the lattice path dataset, underperforming even MLPs and random guessing?
(ii) Can the authors provide concrete examples of new conjectures discovered by ML models that were later validated?

**Relation To Broader Scientific Literature:**

The work builds on existing ML approaches for mathematical reasoning and shifts the focus from theorem proving to conjecturing. This work is also closely related (and potentially contributing to) formal proof verification (e.g., Lean, Coq), and the  Fajtlowicz’s Graffiti conjecture-generating systems.

**Theoretical Claims:**

The paper does not introduce new theoretical results but leverages known results in algebraic combinatorics to structure the datasets.

---

> ### Author Rebuttal · Authors · 2025-04-01
>
> We would like to thank the reviewer for their thoughtful feedback and questions. We especially appreciated the questions about differences in model performance, which we also have. We provide responses to the points in the review below.
>
> - *Addition of examples of conjectures powered by machine learning that were later validated by proofs*
>      - We agree that providing some precedent for this work would be helpful. In the paper we now cite the following. (i) A specialized graph neural network architecture was developed in [1] to explore a problem related to Kazhdan-Lusztig polynomials and insights from this work were used to prove some new theorems in [2]. (ii) In [3] the authors use machine learning to study the geometry of affine Deligne-Lusztig varieties and prove a new lower bound on the dimension of these geometric objects. (iii) We have added a description of some of our own ongoing work where an interpretability analysis of a model trained on a subclass of weaving patterns led to a new theorem.
> - *Differences in architecture performance, especially with respect to lattice paths*
>      - Thanks for noting this. We have added some additional text to Section C which discusses this issue and speculates on some reasons that a given architecture may have underperformed on a dataset.
>           1. **Data representation:** Many of the datatypes (e.g., permutations, partitions, and lattice paths) are rarely used as input data within machine learning research. Consequently, we have a limited understanding of what representations are optimal for a given architecture (there are at least a dozen common representations of permutations in algebraic combinatorics and many look very different). Further, some small-scale experiments we have run suggest (unsurprisingly) that the best representation is partially determined by the solution to the task.
>           2. **Implicit bias or architectural priors:** We speculate that in certain cases architectural priors or implicit bias coming from the training routine may be misaligned with the problem solution. We note on line 1326 that the claimed sensitivity bias of transformers may be at odds with certain problems in combinatorics where the solution requires learning a map that is very sensitive to small changes to input (e.g., permutation parity).
>      - These issues would probably be best explored in smaller, ‘toy’ settings where possible solutions are well-known, rather than the open problems of this collection.
>      - We were also surprised by the inability to obtain good performance with transformers on the lattice paths dataset and had several team members train on these independently to confirm this issue. Currently, our best guess is that because we restrict to covering relations (a small subset of all possible order relations), our dataset ends up being sparsely sampled in terms of the problem space. We have seen transformers underperform in similar settings.
> - *Dataset diversity claim*
>      - The reviewer is correct that this is mostly speculation. We will soften our language.
> - *The computational cost of training models*
>      - We have added a table to the appendix to capture these statistics.
> - *Concatenating the appendix pdf with the main paper*
>      - Unfortunately, ICML does not let us submit these as one document.
> - *Mention of Lean, Coq, and Graffiti*
>      - Thank you for pointing this out, we now reference these.
> - *Some datasets are relatively small, raising concerns about generalizability to more complex mathematical settings.*
>      - We would argue that most of these datasets (other than possibly mHeight) represent complex mathematical settings. The datasets corresponding to open problems must be complex in some sense since mathematicians have made careers out of trying to solve them and yet they remain open. Even the datasets associated with problems that are not open, such as characters of the irreducible representations of the symmetric group (where a combinatorial algorithm has been known for over 70 years) still has many questions associated with it (for instance, when characters are zero or not). In all cases we provide or point to code that allows the user to generate larger datasets for larger $n$.
>      - Finally, while we hope that our datasets will be useful to evaluate the efficacy of ML for math techniques, we stress than a non-generalizing method that helps solve one of these open problems would be a huge accomplishment.
>
> [1] Davies, Alex, et al. "Advancing mathematics by guiding human intuition with AI." Nature 600.7887 (2021): 70-74.
>
> [2] Blundell, Charles, et al. "Towards combinatorial invariance for Kazhdan-Lusztig polynomials." Representation Theory of the American Mathematical Society 26.37 (2022): 1145-1191.
>
> [3] Dong, Bin, et al. "Machine Learning assisted exploration for affine Deligne–Lusztig varieties." Peking Mathematical Journal (2024): 1-50.

---

> > ### Comment · Reviewer_eme9 · 2025-04-05
> >
> > Thanks for your clarification! I raised my rating from 3 to 4.

---

### Official Review · Reviewer_pA6U · 2025-03-08

**Overall Recommendation:** 3

**Summary:**

The paper introduces the Algebraic Combinatorics Dataset Repository (ACD Repo), a collection of nine datasets to use AI in advancing research-level algebraic combinatorics.
A key contribution from author is the dataset focus on open problem with a large collection of examples. A clear focus from author is not just creating a new benchmark (models on some problems can easily do >90%) but more to see if AI can extract insights which finally lead to conjecturing.
The paper provides initial baselines on different datasets using different AI models with or without the language component.

**Claims And Evidence:**

No particular claim has been made.

**Essential References Not Discussed:**

N/A

**Experimental Designs Or Analyses:**

The work includes experimental results of standard architectures to the datasets.

**Methods And Evaluation Criteria:**

Authors are not proposing a new method, but a new dataset.

**Other Comments Or Suggestions:**

N/A

**Other Strengths And Weaknesses:**

Strengths
1. Their focus is really at the conjecturing process, which is a sizable chunk of the research activity.
2. The paper includes examples of how interpretability analysis and LLMs can be used to extract mathematical insight and generate conjectures.

**Questions For Authors:**

1. Is there any particular reason why some problems are not interesting outside a certain boundary? For example, looking at the Github it says that for a particular problem N can be only 6/7/8.
2. There are math areas where even small Ns lead to intractable problems. Can you clarify if this is the case for the area as well?

**Relation To Broader Scientific Literature:**

This is one of the first attempt to propose challenging and open problems for the AI community.

**Theoretical Claims:**

There is no theoretical claim in the paper.

---

> ### Author Rebuttal · Authors · 2025-03-31
>
> We would like to thank the reviewer for the interesting questions. We provide answers below.
> - *Is there any particular reason why some problems are not interesting outside a certain boundary? For example, looking at the Github it says that for a particular problem N can be only 6/7/8.*
>      - This is a good question. In most cases we have provided the code for a user to generate any $n$ they are interested in. Of course, for sufficiently large $n$ it becomes too compute or memory intensive to generate or store the full dataset (storing all permutations of 20 elements is expensive) and at some point, even computing individual instances becomes expensive. In most cases, we decided to provide several datasets that sit at what we would consider the “sweet spot”, large and complex enough to be used to train model ML architectures but small enough that researchers with limited compute budgets could still work with them (this is generally in the 10K to 10M range). We will plan to add this reasoning into a section at the beginning of Section B.
> - *There are math areas where even small Ns lead to intractable problems. Can you clarify if this is the case for the area as well?*
>      - One can easily find problems where data is very hard to generate beyond the trivial size or where specific families of examples are hard to generate. One example of the latter comes from Kazhdan-Lusztig polynomials with $\mu$-coefficient (Section 4.4) which is neither 0 nor 1. This is an area of interest to researchers. Unfortunately, the first such instance appears for $n=10$ at which point there are $(10!)^2$ polynomials (not necessarily distinct). We had initially aimed to include a problem around $\mu$ coefficients but abandoned it because the computational burden was too great.

---

> > ### Comment · Reviewer_pA6U · 2025-04-05
> >
> > Thanks for these answers, I will keep the score!

---

### Official Review · Reviewer_vbej · 2025-03-08

**Overall Recommendation:** 3

**Summary:**

The authors introduce a collection of datasets called the Algebraic Combinatorics Dataset Repository, which contains 9 datasets including an open-ended research question and many examples that should be used to derive conjectures. The authors describe the mathematical background of each dataset as well as results of training some neural models on them. The authors demonstrate how conjectures might be derived based on training machine learning models on this data, and leave the dataset as a good testbed for looking for conjectures.

**Claims And Evidence:**

The authors claim that the use of problems from algebraic combinatorics is a natural choice for their dataset, and I agree with them. The dataset is nicely curated to be useful to researchers with lesser mathematical background, as the problems do not require much background to understand. Although I am not familiar with the particular problems included int he dataset, the authors do a good job motivating their importance/relevance for such a dataset.

**Essential References Not Discussed:**

I think the related works is quite slim in the present form. I think the authors can include at least the two following citations: the Ramanujan Machine (which generated conjectures relating to well known constants like \pi and e) and Graffiti, which generated conjectures in graph theory. I think the authors can also include further references in conjecturing, there are many in the literature. (Notably, what I suggested does not include machine learning, but I still think these are useful to include).

**Experimental Designs Or Analyses:**

The concern I have about experimental design is the same as mentioned in the above section. While the authors train on transformers and MLPs, they don't actually mention whether or h ow these would be useful models/training regimes for extracting conjectures. The GNN based approach, and the program synthesis approach they actually experiment with seem to be much better candidates for demonstrations for researchers on how one may go about producing conjectures.

I think the paper should contain much more information in the main body about the program synthesis experiment.

**Methods And Evaluation Criteria:**

The authors target the field of using machinel earning models on raw mathematical data to make conjectures. I believe that the methods they experiment with make sense, and are generally well motivated.

However, the justification for machine learning methods assisting with making conjectures involves a GNN/XAI based approach, and a program synthesis approach using LLMs. However, the authors train MLPs and transformers on their data and report results stemming from that. This seems to be at contrast with the more explainable methods they cite and use, because they don't give any explanation of how one might use MLPs/transformers to extract conjectures. In this way the experiments they run on the datasets seem to be confused with the broader goal of the dataset and the submission..

**Other Comments Or Suggestions:**

At the header of section 4.2 I think it may be better to leave the (Key Tool in the solution of a recently solved conjecture) in the body of the section, not in the header. My feeling is that this is too verbose.

In section 5, the part about the challenges of making such datasets does not feel to be in the appropriate spot. This may be better included in the limitations of the work.

I also believe that the program synthesis for schubert polynomials, as an example, should appear first in the experiment section. This is because this is an experiment carried out by the authors. The GNNs example including (He et al. 2024) is not a contribution of this paper, and should be secondary to experiments carried out by the authors.

On line 434 in the conclusion, I recommend changing `can't` to aovid using contractions.

Also, I think you may mention FunSearch (Deepmind, '23) in your discussion of your program synthesis based approach - that methodology may be successfully applied to find conjectures fitting the datasets.

**Other Strengths And Weaknesses:**

-

**Questions For Authors:**

1. Can the authors clarify the purpose of evaluating MLPs and transformers on each of the problems? Would it not be better to use some other more interpretable methods? (I do understand you don't want to do the work for people who would use your dataset! But how might one actually use MLPs and transformers to create conjectures).
2. By what process did you select these problems for your dataset? This information should also be included in the paper for enabling others to find suitable problems for further exploration.

**Relation To Broader Scientific Literature:**

I think the submission has a nice spot in the broader scientific literature for AI-for-math. At present there have not been (to my knowledge) datasets of this kind that support mathematical conjecturing from raw data. This is probably in part due to the AI-for-math research having a larger community presence of AI researchers, who are less familiar with mathematical results than the researchers from the maths world. I think this dataset provides a good starting point for researchers interested in this area.

**Theoretical Claims:**

There are no theoretical claims in this submission.

---

> ### Author Rebuttal · Authors · 2025-03-31
>
> We would like to thank the reviewer for providing feedback on the paper, especially for pointing out that we may need to provide more motivation for our choice of baselines and for suggestions on the paper’s structure. We provide responses to the points in the review below.
> - *Tension between baselining on MLPs and transformers while only providing example approaches that use more sophisticated methods.*
>      - Thank you for bringing this up. We agree that we may not have effectively communicated the purpose of our baselines. Most approaches to extracting mathematical insight from an ML model first require one to have a model that is performant on some task related to the problem one cares about. Of course, this is just a necessary but not sufficient condition (the second step, extracting insights seems to be harder in our experience). Many potential users of these datasets (especially those coming more from the mathematics community) will have some experience and intuition for MLPs and transformers. Far fewer will have experience with GNNs or program synthesis paradigms. We hope that the baselines we provide will give the broadest possible audience a sense of how "hard" it is just to get a performant model (acknowledging that problem "hardness" ultimately depends on the method used). We also did not want to bias users toward only applying program synthesis approaches which, though powerful, also have limitations. We have added in a short section at the beginning of Section 4 clarifying the purpose of the baselines.
>      - This all being said, we also recognize that program synthesis will be a topic of special interest at ICML this year so we will plan to move some additional program synthesis details back into the main body of the paper in our next revision.
>      - Finally, we agree with the reviewer that adding a third example in Section 5 that uses either MLPs or transformers would be appropriate given their central place in the baselines. We have included a short description of some results we obtained on a subclass of weaving patterns. We obtained these by using clustering to identify prototypical patterns in the Shapley values associated with a performant model. Analysis of these prototypes guided us to a formal characterization of the matrices corresponding to this subclass.
> - *Program synthesis results first in Section 5*
>      - This is a good point! We have now made this change.
> - *Mention of FunSearch*
>      - Omitting this was an oversight. We have added it in, thanks!
> - *Essential references*
>      - Thank you for pointing these out! We have added these.
> - *By what process did you select these problems for your dataset?*
>      - In choosing questions we aimed to have decent coverage of the major areas of research in algebraic combinatorics. One way of doing this is to ensure that the major combinatorial gadgets (e.g., permutations, Young tableaux, etc.) are featured. Algebraic combinatorics draws from different subfields of algebra, for instance representation theory or algebraic geometry. We also tried to make sure we had some coverage from this angle as well. It is easy to make up problems which are open but which no one cares about. To avoid this, all of our open problems were suggested by two experts in the field. The machine learning researchers on the team then checked that these problems seemed reasonable for application of machine learning and then reformulated them in an ML-friendly format. We have now added several sentences at the beginning of Section 4 addressing this.
> - *Header on Section 4.2*
>      - Thank you, we have changed this.
> - *Location of challenges section*
>      - We agree with this comment and have made this change.
> - *Use of 'can't'*
>      - We removed the contraction, thanks!

---

> > ### Comment · Reviewer_vbej · 2025-04-02
> >
> > Thank you for the detailed reply to my questions and notes! I agree that a third example using MLPs/Transformers in Section 5 (perhaps as the first, or second in order) would be very useful for this paper. It also was helpful to hear about how you chose these problems, thanks! I will keep my score.

---

### Official Review · Reviewer_zx12 · 2025-03-14

**Overall Recommendation:** 4

**Summary:**

The paper introduces nine datasets arising from problems in algebraic combinatorics. These datasets are meant to test capabilities of machine learning models on symbolic math tasks. The authors clearly motivate and describe each of the nine problems and the resulting benchmarks. Several baseline methods are tested on the benchmarks. Useful discussion is provided showing how development of ML models for these benchmarks can lead to progress in mathematics.

**Update after rebuttal**

The authors sufficiently addressed my objections, so I raised my score and I feel I can recommend the paper for acceptance.

**Claims And Evidence:**

The main contribution of the paper is the introduction of a novel benchmark, so the focus is on presenting the benchmark and not on stating and verifying claims. However, the authors want to show that the datasets are challenging for standard ML approaches, and to this end baseline experiments are conducted. I'm not completely sure how to interpret the results -- in fact, it seems that for the majority of the datasets the baseline methods (especially MLP) could achieve quite high performance.

**Essential References Not Discussed:**

I think the authors in general include relevant references, however, it would be perhaps good to discuss [1], where also synthetic math benchmarks for ML are created, and [2], where the studied ML task is to generate formulas describing integer sequences from OEIS.

[1] Saxton et al.: Analysing Mathematical Reasoning Abilities of Neural Models. ICLR 2019\
[2] Gauthier, Urban: Learning Program Synthesis for Integer Sequences from Scratch. AAAI 2023

**Experimental Designs Or Analyses:**

I didn't see any issues with the experimental design.

**Methods And Evaluation Criteria:**

The authors in general choose appropriate methods and evaluation criteria for providing the baselines.

**Other Comments Or Suggestions:**

It would be better to provide appendix in the same pdf as in the main text -- now some links (e.g. Table 3) do not work.

In Table 2, in the caption you mention Claude 3.5 Sonnet but in the table itself it is not present.

The captions of the tables should be placed above, per formatting instructions.

"this example comes from (war)" -- this reference needs fixing

Table Table --> Table
these these --> these
Machine Learning meets --> Machine Learning Meets

**Other Strengths And Weaknesses:**

### Strengths

The paper introduces a novel and interesting benchmark which connects ML research with more advanced mathematics. In my opinion, such benchmarks are highly desirable, and may stimulate developments on both ML and mathematics side.

The math problems and resulting data are well-described and motivated, making them accessible for non-experts.

### Weaknesses

Some technical details related to training the baselines are missing, for instance:
- What training hyperparameters were used (n. of epochs)?
- What architectural hyperparameters were used (n. of layers)?
- How logistic regression was optimized?

Most of the problems can be solved with the baseline methods with quite high accuracy, which makes one wonder how difficult the benchmarks actually are.

### Conclusion
In general, I find the work very useful, and once the authors resolve the issues raised (the problems with the code, missing technical details), I'm willing to raise the score.

**Questions For Authors:**

1. I see that MLP in general performs better than the transformer, but sometimes it's the other way around (e.g., on "cluster algebra quivers"). Do you maybe have a hypothesis why?
2. What are the numbers presented in Table 3? is it mean squared error?
3. I didn't understand how the RSK problem is cast as a regression task -- could you elaborate on that?

**Relation To Broader Scientific Literature:**

This work is related to the literature on applying ML to math problems, and more specifically, to math benchmarks. There are several categories of existing math benchmarks: formal mathematics benchmarks, natural language math benchmarks of varying difficulty, and synthetic math problems. This work falls in the last category, and because there are not many such benchmarks, I find the contribution of the authors very useful.

**Theoretical Claims:**

The authors introduce a bit of background theory in algebraic combinatorics to motivate the problems resulting in the datasets. I'm not familiar with this domain of mathematics, so I myself wasn't able to judge how appropriate the problems are -- but I believe the problems indeed are well-motivated and appropriate, as the authors seem to be well-versed in algebraic combinatorics.

---

> ### Author Rebuttal · Authors · 2025-03-30
>
> We would like to thank the reviewer for all their thoughtful comments on the paper and for pointing out issues with the GitHub page. We provide responses to the points in the review below.
> - *“…it seems that for the majority of the datasets the baseline methods … could achieve quite high performance.”*
>      - This is a good point and one that we tried to address in lines 086-091. Our goal is a little different than a standard benchmark where model performance alone is what we care about (this is one reason we ultimately decided to remove the word 'benchmark' from the paper). When looking for mathematical insight, being able to train a model to perform the task associated with an open problem is a necessary, but not sufficient condition. We must also be able to extract mathematical insight from the model. The model that has just learned a ‘bag of heuristics’ might be highly performant but would probably be hard to extract clean theorems from. The inclusion of the benchmarks was meant to give the reader a better sense of how challenging the first step is (getting a performant model). We have now clarified this in the text.
> - *Installation instructions*
>      - Thank you for pointing this out. These have now been added to the [GitHub repository](https://github.com/icml2025-43403439/43403439/blob/main/README.md#environment-installation).
> - *Java program*
>      - This was code written by one of our mathematician collaborators for their research prior to the inception of the ACD repo. The Java version has now been replaced with a Python version.
> - *Errors when loading data*
>      - Thank you for pointing these out. We now believe we have fixed all the issues listed.
> - *One link to download all datasets*
>      - This is a good idea! We now have a single script (`load_datasets.py`) which can be run to download, all datasets, unzip them and move them to the correct folder (other than KL polynomials, RSK, and Schubert, which are too large to download programmatically via the API).
> - *Additional references*
>      - Good catch! We have added both.
> - *Training and architectural hyperparameters*
>      - The range of hyperparameters (including number of layers, depth, etc.) that we explored can be found in Section C.1 of the Supp. Material. We will add a table with the specific hyperparameters we used for each dataset (an abbreviated copy of the transformers table can be found below). We have also added training epochs (60), batch size (varied between datasets), loss function (cross-entropy for classification and MSE for regression), and Adam hyperparameters (the Pytorch default other than learning rate) into Section C.1.
> | Dataset | LR | Depth | Dimension | Heads |
> | -| - | - | - | - |
> | Lattice, $n=10$ | 0.001 | 4 | 80 | 8 |
> | Lattice, $n=11$ | 0.0005 | 4 | 80 | 4 |
> | Lattice, $n=12$ | 0.0005 | 6 | 40 | 8 |
> | Weaving, $n = 6$ | 0.0001 | 4 | 80 | 8 |
> | Weaving, $n = 7$ | 0.0001 | 4 | 80 | 8 |
> | Quivers | 0.0005 | 6 | 80 | 4 |
> | Grassmannian | 0.0005 | 6 | 80 | 4 |
> | Schubert $n = 4$ | 0.0001 | 4 | 80 | 6 |
> | Schubert $n = 5$ | 0.0005 | 4 | 40 | 6 |
> | Schubert $n = 6$ | 0.0005 | 4 | 80 | 4 |
> | mHeight, $n = 8$ | 0.0001 | 4 | 80 | 4 |
> | mHeight, $n = 9$ | 0.001 |  6 | 20 | 8 |
> | mHeight, $n = 10$ | 0.001 | 4 | 80 | 8 |
> | $S_n$, $n = 18$ | 0.001 | 6 | 80 | 6 |
> | $S_n$, $n = 20$ | 0.001 | 4 | 80 | 8 |
> | $S_n$, $n = 22$ | 0.001 | 4 | 40 | 8 |
> | RSK, $n = 8$ | 0.001 | 4 | 20 | 6 |
> | RSK, $n = 9$ | 0.001 | 4 | 80 | 4 |
> - *Logistic regression optimization?*
>      - A short description is provided in Section C.1. We used the default settings for logistic regression provided by sklearn (we have now replaced the word 'standard' by 'default').
> - *MLPs vs. transformer performance*
>      - This is a great question to which we mostly don’t have answers (especially for open problems where the solution is unknown). We see the choice of data representation as one critical unknown for many of these datasets. For example, what is the best way to represent a permutation when using a transformer vs MLP? We have also seen evidence that the transformer's sensitivity bias limit it in certain math settings (e.g. parity problems).
> - Typos
>      - These have now all been addressed, outside of the supplementary material being split from the main body. Unfortunately, ICML does not let us submit these as a single document.
> - *The meaning of Table 3*
>      - Correct. We have added a note to clarify this.
> - *RSK regression task*
>      - Permutations are represented as their inversion vector, a $\{0,1\}$ vector whose first entry is 1 if the permutation inverts the ordering of 1 and 2, whose second entry is 1 if the permutation inverts the ordering of 1 and 3, etc. This vector is the target of regression. Doubtless better frameworks could be devised by one interested in this task, but that was not our goal in the baselines. Clarification has been added to Section B.6.

---

> > ### Comment · Reviewer_zx12 · 2025-04-05
> >
> > Thank you for the rebuttal and fixes in the code! I'm increasing my score, and have two additional remarks:
> >
> > (1) Thank you for the `load_datasets.py` script, it seems to work, but not for all datasets. For example, for RSK I see:
> > ```
> > Downloading rsk from Google Drive (file_id=1CfuxD_XgTefbEduxJnXgXoUOt-GY-smq) to datasets/rsk.zip...
> > Finished downloading rsk.
> > Unzipping rsk.zip into data...
> > [ERROR] rsk.zip is not a valid zip file - you will need to download the data manually.
> > ```
> > Could you have a look what happens here?
> >
> > (2)
> > > Permutations are represented as their inversion vector [...] This vector is the target of regression.
> > Regression predicts a continuous variable, so is this binary vector cast as a continuous variable somehow?

---

> > > ### Author Response · Authors · 2025-04-05
> > >
> > > Thank you for your additional remarks!
> > >
> > > (1) *The `load_datasets.py` script does not pull all the datasets*.
> > >
> > > For large files (> 150MB), Google Drive prompts for manual confirmation, so `load_datasets.py` cannot easily pull them programmatically. To avoid having to manually download the RSK/Schubert/KL datasets, we have collected [all the data into a new link here](https://drive.google.com/file/d/1A5DlXHj81c5JlgpuCoS5FtMahbSHAyyj/view), and have updated the README.md to describe both approaches for pulling the datasets:
> > > > ### Downloading the datasets
> > > > You have two options for downloading the datasets.
> > > >
> > > > 1. Manually via a Google Drive link
> > > >
> > > > The datasets can be downloaded here: [https://drive.google.com/file/d/1eWcXsNPAsCJMsVqcoYscUUz9S0-wjKnY/view?usp=sharing](https://drive.google.com/file/d/1A5DlXHj81c5JlgpuCoS5FtMahbSHAyyj/view).
> > > > After downloading the file, unzip it into a folder called `data` (assuming you have put the downloaded zip file in the same directory as this README):
> > > > ```bash
> > > > unzip all_data.zip -d data
> > > > ```
> > > >
> > > > 2. Programmatically via the Google Drive API (grabs 6/9 datasets)
> > > >
> > > > To download the data into a folder called `data` run:
> > > > ```bash
> > > > python load_datasets.py
> > > > ```
> > > > Note however that the datasets for KL polynomials, RSK, and Schubert polynomials are too large to download programmatically via the Google Drive API and have to be downloaded manually. See their respective README files for more details.
> > >
> > > (2) *Representing the RSK task as a regression problem by casting the binary vector*.
> > >
> > > That is correct. What we are calling the inversion vector is a vector of 0’s and 1’s which we convert to floats and treat as a vector with continuous values. MSE loss is then computed across entries in the vectors.

---

### Decision · Program_Chairs · 2025-05-01

**Decision:**

Accept (oral)

**Comment:**

This paper introduced a suite of benchmark problems that captures research-level mathematics. Current benchmarks in AI4Math focus almost exclusively on pre-college math such as IMO. The proposed benchmark fill in this gap and is a valuable contribution to the community. The paper received unanimous accepts from reviewers. The authors are encouraged to incorporate the reviewers' suggestions in the camera-ready version.